# Dynamic root microbiome sustains soybean productivity under unbalanced fertilization

Mingxing Wang[1,2,9], An-Hui Ge[1,9], Xingzhu Ma[3,9], Xiaolin Wang[4], Qiujin Xie[1], Like Wang[1,2], Xianwei Song [5], Mengchen Jiang[1], Weibing Yang[1], Jeremy D. Murray[1], Yayu Wang [6], Huan Liu [6,7], Xiaofeng Cao [5] ✉ & Ertao Wang [1,8] ✉

Root-associated microbiomes contribute to plant growth and health, and are dynamically affected by plant development and changes in the soil environment. However, how different fertilizer regimes affect quantitative changes in microbial assembly to effect plant growth remains obscure. Here, we explore the temporal dynamics of the root-associated bacteria of soybean using quantitative microbiome profiling (QMP) to examine its response to unbalanced fertilizer treatments (i.e., lacking either N, P or K) and its role in sustaining plant growth after four decades of unbalanced fertilization. We show that the root-associated bacteria exhibit strong succession during plant development, and bacterial loads largely increase at later stages, particularly for Bacteroidetes. Unbalanced fertilization has a significant effect on the assembly of the soybean rhizosphere bacteria, and in the absence of N fertilizer the bacterial community diverges from that of fertilized plants, while lacking P fertilizer impedes the total load and turnover of rhizosphere bacteria. Importantly, a SynCom derived from the low-nitrogen-enriched cluster is capable of stimulating plant growth, corresponding with the stabilized soybean productivity in the absence of N fertilizer. These findings provide new insights in the quantitative dynamics of the root-associated microbiome and highlight a key ecological cluster with prospects for sustainable agricultural management.

Root-associated microbiomes have been recognized as the "second genome" of plants that contributes in various ways to plant growth, development and health[1]. The root-associated compartments, including the rhizosphere and endosphere, provide unique habitats for microbial colonization, resulting in substantial taxonomic and functional differences compared with surrounding bulk soils. Root-associated microbiomes are highly dynamic, and strongly affected by plant development[2–4], which exerts its effect mostly through the impact of root exudates on microbial growth[5], indicating the ability of plants to actively modify their microbiomes throughout entire life-cycle. This character may have important implications for plant nutrient acquisition, as evidenced by the temporal complementarity of

[1]New Cornerstone Science Laboratory, National Key Laboratory of Plant Molecular Genetics, CAS Center for Excellence in Molecular Plant Sciences, Institute of Plant Physiology and Ecology, Chinese Academy of Sciences, Shanghai 200032, China. [2]University of Chinese Academy of Sciences, Beijing 100049, China. [3]Heilongjiang Academy of Black Soil Conservation and Utilization, Harbin 150086, China. [4]College of Agriculture, South China Agricultural University, Guangzhou 510642, China. [5]Institute of Genetics and Developmental Biology, Chinese Academy of Sciences, Beijing 100101, China. [6]State Key Laboratory of Agricultural Genomics, Key Laboratory of Genomics, Ministry of Agriculture, BGI Research, Shenzhen 518083, China. [7]BGI Life Science Joint Research Center, Northeast Forestry University, Harbin 150040, China. [8]School of Life Science and Technology, ShanghaiTech University, Shanghai 201210, China. [9]These authors contributed equally: Mingxing Wang, An-Hui Ge, Xingzhu Ma. ✉e-mail: xfcao@genetics.ac.cn; etwang@cemps.ac.cn

nitrogen use efficiency between roots and microbes (e.g., AMF) in wheat[6]. Currently, relative microbiome profiling (RMP) has been widely used to detect variation in the relative abundance of taxa in complex microbial communities, but RMP fails to provide information on the absolute abundance of microbes and is not useful for comparing microbial loads among samples[7-9]. Increasing evidence indicates that the specific load of microbial groups, as estimated by quantitative microbiome profiling (QMP), responds sensitively to environmental disturbance, even when the relative abundance of microbes remains the same[8,9]. However, due to the technical challenges posed by QMP, there remains a lack of detailed high resolution information on the quantitative dynamics of root-associated microbiomes. Thus, data from QMP is urgently needed to advance our understanding of the relationship between root-associated microbiome assembly with plant development.

Apart from host genotype and developmental stage, the assembly of root-associated microbiome is also affected by environmental factors. For instance, drought reduced the diversity and disrupted the temporal dynamics of the root microbiomes of different plant species[10,11]. In agriculture, the utilization of chemical fertilizers, including nitrogen (N), phosphorous (P) and potassium (K), has contributed greatly to yield increases in recent decades, making it possible to feed the expanding human population. However, intensified agriculture not only causes severe environmental pollution, but also inactivates positive plant-microbial interactions, leading to a cascade effect of damage on ecosystem integrity and health[12,13]. N amendment is associated with reduced soil microbial biomass[14,15], and fertilization was found to significantly decrease the microbial temporal succession rate in soil[16]. However, quantitative studies assessing how soil nutrient conditions affect the temporal dynamics of the root-associated microbiome, particularly by the QMP method, are lacking. Understanding how root microbial dynamics respond to different fertilization regimes and affect crop productivity is essential for the development of sustainable agriculture.

Plants are thought to subtly manipulate their root-associated microbiome under nutrient shortage, since most of terrestrial ecosystems facing N and P limitation[17]. A recent study revealed that maize plants recruit Oxalobacteraceae to improve their performance under low N[18], providing strong evidence for microbiome effects on plant growth. In contrast to cereals, legumes, such as soybean, pea and common bean, can form a mutualistic relationship with N-fixing bacteria which accumulate inside nodules and provide a large amount of N for host growth. However, the benefits of nodulation are substantially reduced under N fertilization[19,20], indicating a tradeoff between minimizing plant carbon usage and maximizing the acquisition of fixed N. Having access to their own N supply, legume crop production may instead be limited by P and K inputs, which have been shown to contribute to symbiotic N fixation[20]. Access to these nutrients in plants can be bolstered by its rhizosphere microbiome, which was shown to specifically enrich mineral nutrient metabolism in comparison to bulk soil by microbial priming effects mediated with root exudates[21,22], but whether and how specific microbes can compensate for soil nutrient deficiency for plant growth remains elusive. Uncovering and validating the functions of key microbes in the alleviation of plant nutrient stress is of vital importance for sustainable agroecosystem management.

In this study, a field trail was conducted to illustrate the effect of nutrient deficiency (i.e., exclusion of N, P, or K fertilizers) on soybean performance and the development of the root-associated bacterial microbiome, and to unveil the mechanisms of the root-associated microbiome in conditioning plant phenotype, after 40 years of soybean-maize-wheat rotation. Through QMP, we monitored the dynamics of rhizosphere and endosphere bacterial communities across the lifespan of soybean, and deciphered effects on specific bacterial taxa in response to different fertilization treatments. We then combined metagenomic sequencing to explore the functional adaptation of root-associated bacteria in response to nutrient deficiency. Finally, we validated the plant-growth-promoting functions of synthetic communities (SynCom) in the low-nitrogen-enriched microbial ecological cluster using a cultivation-dependent method. We hypothesize that (1) the QMP would reveal a distinctive dynamic pattern in root-associated microbiome assembly; (2) the dynamics of root-associated bacteria would be largely affected by the lack of N, P, or K fertilizers; (3) functional adaptation of the rhizosphere bacteria under nutrient deficiency would benefit soybean growth.

## Results
### Effects of fertilization on soil chemical properties and soybean performance
To unveil the quantitative assembly of root-associated microbiomes in soybean, we investigated microbiome dynamics across plant developmental stages (1, 4, 7, 14, 28, 42, 60, and 72 days after germination) and their response to fertilization. To do this, we studied soybean plants grown in the field using four unbalanced fertilization treatments, which have been part of an ongoing crop rotation with maize and wheat since 1979, including (1) full-dose fertilization (NPK fertilizer, Control), (2) lack of nitrogen fertilizer (-N), (3) lack of phosphorous fertilizer (-P), and (4) lack of potassium fertilizer (-K) (Fig. 1A). Bulk soil was collected from each plot to monitor soil chemical properties before seed sowing in 2020. Compared with Control, the content of soil alkaline hydrolyzable nitrogen (AHN), available phosphorous and available potassium was reduced by 35%, 95% and 61%, to 112 mg kg$^{-1}$, 7 mg kg$^{-1}$ and 146 mg kg$^{-1}$, in -N, -P and -K treatments, respectively ($P < 0.01$, Fig. 1B), indicating that P was the most severely reduced of the managed nutrients. Soil pH was higher by 1.1 units and soil dissolved organic carbon (DOC) was lower by 5.6 mg kg$^{-1}$ in the -N treatment compared with the Control ($P < 0.001$), whereas soil organic matter (SOM) showed no significant difference among treatments (Supplementary Fig. 1), suggesting a high resilience of SOM across different fertilization regimes.

Interestingly, we found that the soybean yield in the field exhibited no significant difference between Control and -N treatment, and even showed slight increment by 9% across two soybean planting seasons (i.e., 2020 and 2017, Fig. 1C). By contrast, soybean yield was consistently reduced by 25-29% in the -P treatment compared with Control ($P < 0.05$). Meanwhile, the -N treatment significantly increased the number and diameter of root nodules compared to the Control ($P < 0.05$), whereas the -P treatment showed the opposite trend ($P < 0.01$, Supplementary Fig. 2).

### Changes in root-associated microbial diversity across plant developmental stages
Quantitative microbial profiling (QMP) was applied to investigate the dynamics of root-associated bacterial diversity, load, and composition across plant developmental stages and their response to fertilization. The root compartment, plant developmental stage, and fertilization treatment exhibited significant effects on bacterial α-diversity ($P < 0.001$), with consistently higher α-diversity in the rhizosphere than the endosphere (Fig. 2, Supplementary Table 1). In the rhizosphere, the bacterial α-diversity decreased with plant development, especially from days 42 to 72 (Fig. 2B). Compared with Control, the -N treatment exhibited a higher α-diversity at early developmental stages (i.e., days 4, 7, and 14), whereas -P treatment increased α-diversity at later developmental stages (i.e., days 42, 60, and 72) in the rhizosphere ($P < 0.05$), resulting a stable α-diversity with plant development (Fig. 2B). By contrast, the α-diversity in the endosphere was less sensitive to fertilization, reaching its minimum at day 14 and remaining relatively stable in subsequent stages ($P < 0.05$, Fig. 2C), which corresponds to the observed onset of root nodulation in the field at 14 days after germination, indicating the symbiosis might drastically reduce endosphere microbial diversity.

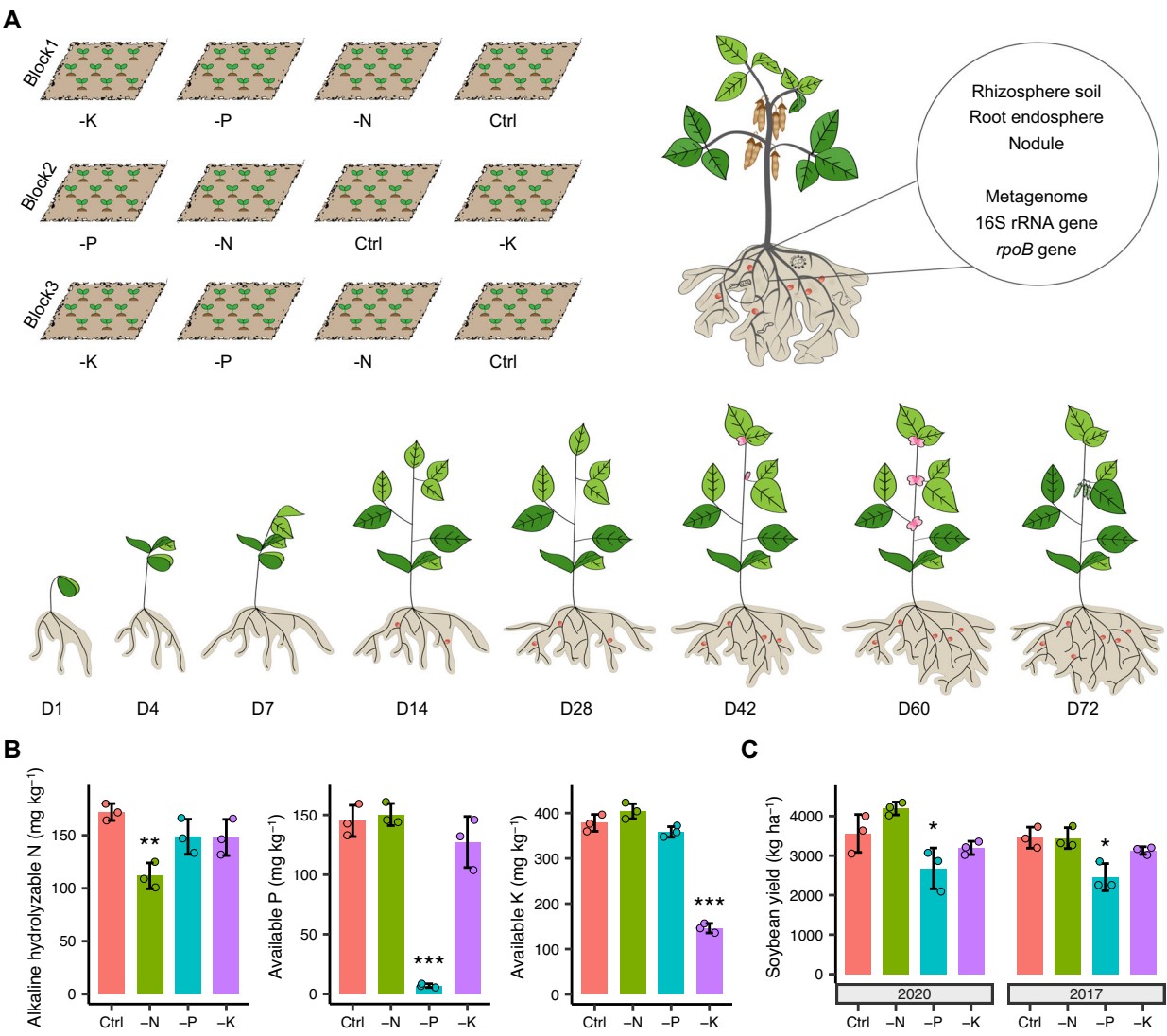

**Fig. 1 | The experimental design and the responses of soil mineral nutrients and soybean yield to unbalanced fertilization. A** the field experiment was arranged in a completely randomized block design. Four treatments, including complete fertilization (NPK fertilizer, Control) and a lack of N, P, or K fertilizer (i.e., -N, -P, and -K treatments), have been applied as part of a soybean-maize-wheat rotation (one crop/year) since 1979. The root-associated compartments of soybean, including the rhizosphere soil, root endosphere, and nodules, were subjected to 16S rRNA gene, *rpoB* gene, and metagenomic sequencing across plant developmental stages. **B** chemical properties of the bulk soils in different fertilization treatments before soybean planting in 2020 ($n = 3$ plots). **C** effect of unbalanced fertilization on soybean yield in 2020 and 2017 ($n = 3$ plots). The asterisks represent the level of significance (*$P < 0.05$, **$P < 0.01$, ***$P < 0.001$) between Control and unbalanced fertilization treatments based on one-way ANOVA test with Dunnett's post hoc analysis (for data fit normal distributions and homogeneous variance) or Kruskal-Wallis test with Dunn's post hoc analysis (for data does not fit normal distributions or homogeneous variance). Exact $P$-values are listed in the Source Data file. The data are presented as mean values ± standard deviation (SD). Source data are provided as a Source Data file.

Principle coordinate analysis (PCoA) based on Bray-Curtis distance of QMP datasets revealed that the bacterial β-diversity was clearly separated among bulk soil, rhizosphere soil and root endosphere (Supplementary Fig. 3). Correspondingly, PERMANOVA results suggested that the root compartment (rhizosphere vs. endosphere) was the main driver of bacterial β-diversity ($R^2 = 0.241$, $P < 0.001$), followed by the plant developmental stage ($R^2 = 0.114$, $P < 0.001$) and fertilization treatment ($R^2 = 0.040$, $P < 0.001$, Supplementary Table 2). For both the rhizosphere and endosphere, the bacterial β-diversity was clearly associated with the plant developmental stage in PCo1 axis (Fig. 2E, F). However, the -N treatment grouped separately from other treatments in the rhizosphere and bulk soils, but not in the endosphere (Fig. 2D–F). Similarly, fertilization had much higher effect on bacterial β-diversity in the rhizosphere than the endosphere, irrespective of the plant developmental stage (Fig. 2E, F, Supplementary Fig. 3), suggesting the rhizosphere microbiome is relatively responsive to fertilization. The temporal-decay pattern of the rhizosphere microbiome revealed that the -P treatment had a significantly lower regression slope with plant development at 0.0014 d$^{-1}$ compared with 0.0043 d$^{-1}$ in the Control, but remained similar in other treatments, while the root endosphere had a relatively low microbial turnover rate ranging from 0.0024 to 0.0031 d$^{-1}$ (Fig. 2G, Supplementary Table 3). The Bray-Curtis distance between the Control and -N treatment in the rhizosphere was relative stable across plant development (Slope = 0.0005 d$^{-1}$), whereas the -P and -K treatments showed a sharp increase of dissimilarity with time compared with Control (Fig. 2H, Supplementary Table 4), indicating the microbial communities in the -P and -K treatments gradually diverged to the Control with plant development. Although similar trends were observed in the microbial α-diversity based on the relative microbial profiling (RMP), the compartment effect of microbial β-diversity was specifically reduced ($R^2 = 0.136$, $P < 0.001$) in comparison to QMP, and the fertilization

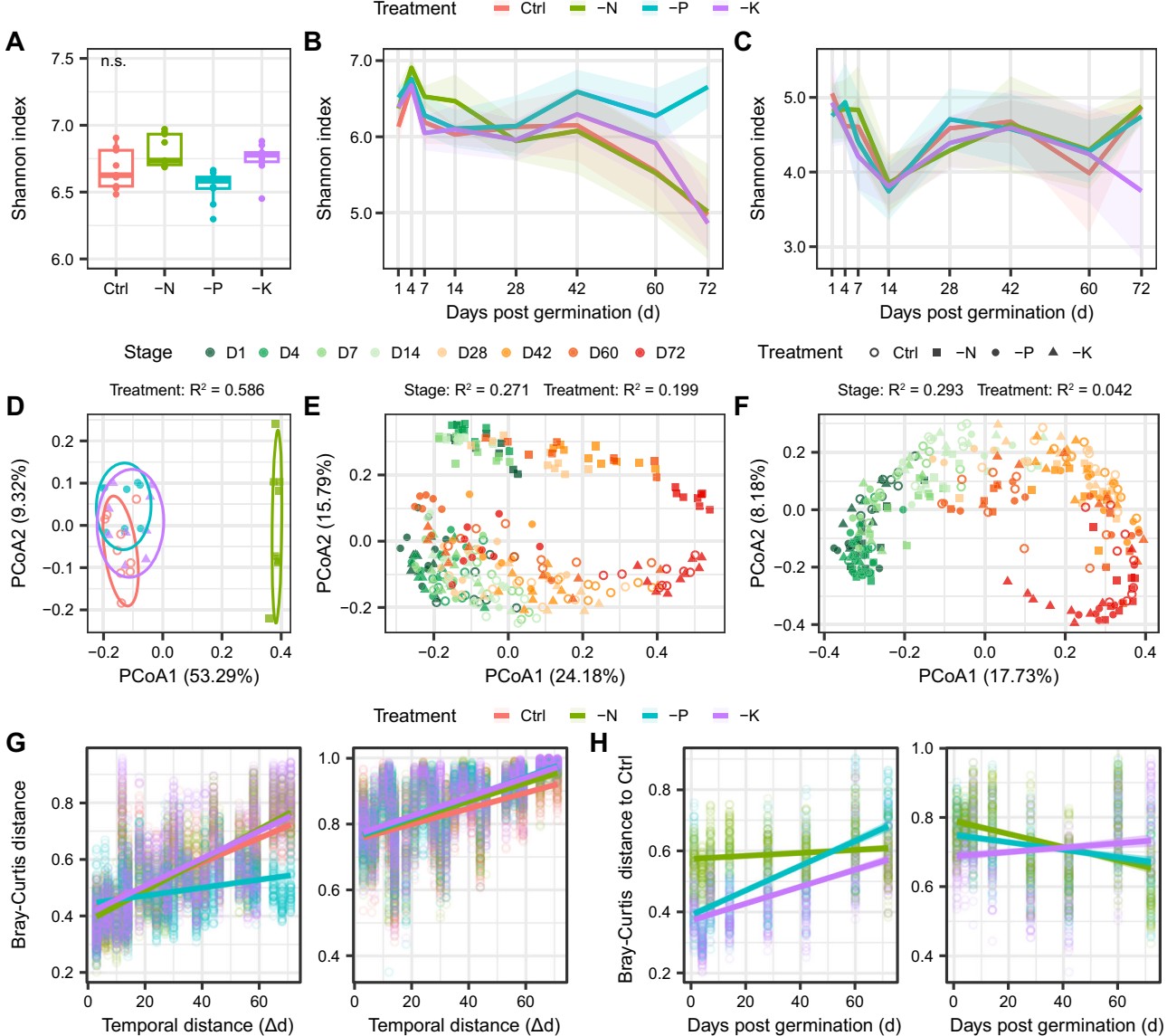

**Fig. 2 | Temporal dynamics of root-associated bacterial diversity in different treatments by quantitative microbiome profiling (QMP). A** bacterial α-diversity (Shannon index) in the bulk soils of each treatment (*n* = 9 soil samples). The box plots indicate the median (center line), the 25th and 75th percentiles (box), and the range of non-outlier values (whiskers). The "n.s." represents the non-significance of the Shannon index between Control and unbalanced fertilization treatments by Kruskal-Wallis test with Dunn's post hoc analysis. Dynamics of bacterial α-diversity in different treatments across plant developmental stages in the rhizosphere (**B**) and root endosphere (**C**). Error bands show the standard deviation (SD). **D** bacterial

β-diversity in the bulk soils analyzed using principle coordinate analysis (PCoA). The symbol colors in (**D**) are consistent with legend in (**A**–**C**). **E** PCoA of bacterial β-diversity in the rhizosphere. **F** PCoA of bacterial β-diversity in the endosphere. **G** linear regressions between temporal distance (change of sampling day between each two samples, Δd) and Bray-Curtis distance among samples in each treatment in the rhizosphere (left) and endosphere (right). **H** linear regressions between sampling stage (days post germination, d) and Bray-Curtis distance of each unbalanced fertilization treatment to the Control in the rhizosphere (left) and endosphere (right). Source data are provided as a Source Data file.

effect was comparable between the rhizosphere and endosphere across plant developmental stages, with relatively lower succession rates in all treatments in the rhizosphere (Supplementary Fig. 4, Supplementary Table 3). Specifically, we observed that 38.1–48.3% of ASVs in the rhizosphere exhibited a distinctive pattern across plant development between QMP and RMP datasets (Supplementary Fig. 5), suggesting a substantial bias when interpreting microbial data with relative abundance.

## Assembly of root-associated microbiome during plant development

The bacterial loads in both the rhizosphere and endosphere exhibited increasing trends with plant development, and bacterial abundances

were comparable across days 1–14 ($5.8 \times 10^9$ copies g$^{-1}$ in the rhizosphere and $1.5 \times 10^8$ copies g$^{-1}$ in the endosphere), but gradually increased and reached their highest abundance at day 72 ($2.3 \times 10^{10}$ copies g$^{-1}$ in the rhizosphere and $4.7 \times 10^9$ copies g$^{-1}$ in the endosphere) (Fig. 3, Supplementary Fig. 6). Fertilization treatments had a significant effect on bacterial loads (Supplementary Table 5), and the microbial load was found to be consistently reduced in the rhizosphere of the -P treatment compared to the Control, especially at later developmental stages, showing bacterial load reductions of 54%, 61% and 75% at days 42, 60, and 72, respectively ($P < 0.05$, Fig. 3B). By contrast, the change of bacterial loads between Control and -P treatment in the endosphere was hardly observed and not consistent during plant developmental stages (Fig. 3C).

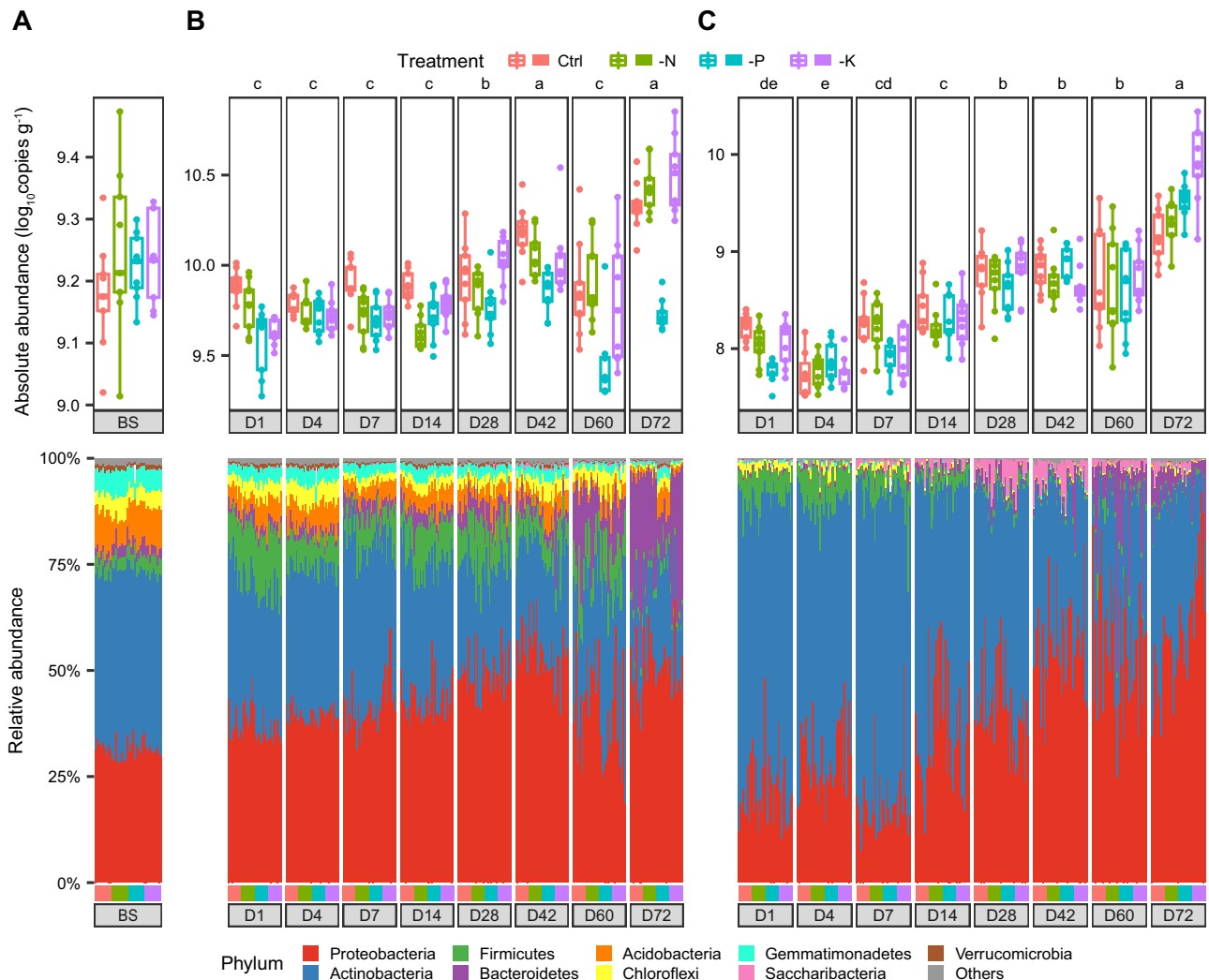

**Fig. 3 | Bacterial loads and composition in the bulk soil, rhizosphere and endosphere based on 16S rRNA sequencing data. A** absolute abundance and bacterial community composition at the phylum level in the bulk soil (*n* = 9 soil samples). Temporal dynamics of bacterial abundance and community composition in the rhizosphere (**B**) and root endosphere (**C**). The color in the upper panel represents different treatments, and the color in the lower panel represents bacterial phyla, as indicated in the accompanying legends. Different letters indicate

significant difference of the average bacterial load among developmental stages at *P* < 0.05 by Kruskal-Wallis test with Dunn's post hoc analysis for multiple comparisons. Sample size, replicates and exact *P*-values are listed in the Source Data file. The box plots indicate the median (center line), the 25th and 75th percentiles (box), and the range of non-outlier values (whiskers). Source data are provided as a Source Data file.

The root-associated bacteria mainly belonged to the Proteobacteria, Actinobacteria, and Bacteroidetes, with Actinobacteria being more dominant at early stages and Proteobacteria being more dominant at later stages, particularly in the endosphere (Fig. 3, Supplementary Fig. 6). Although the relative abundance of Actinobacteria exhibited a progressively decreasing trend with plant development (reducing from 33.4% and 68.3% at day 1 to 18.0% and 32.6% at day 72 in the rhizosphere and endosphere, respectively), their absolute abundances increased by 2.1 and 18.8 fold in the rhizosphere and endosphere, respectively (Figs. 3, 4), indicating that the plants support an increasing load of Actinobacteria despite their reduced relative abundance. Meanwhile, the number of Proteobacteria increased by 5.4 and 119.2 times during plant developmental stages in the rhizosphere and endosphere, respectively (Fig. 4). Intriguingly, the Bacteroidetes were hardly detected at early stages (2.3% and 0.5% of relative abundance at day 1 in the rhizosphere and endosphere), but transitioned to rapid growth at day 60 and multiplied by 48.8 and 460.7 fold at day 72 relative to day 1 in the rhizosphere and endosphere, respectively, with 23.2% and 5.7% of relative abundance (Figs. 3, 4).

## Root-associated microbiome in response to fertilization

Fertilization treatments had a significant effect on the composition and dynamics of the root-associated microbiome (Figs. 2, 3). At the phylum level, the abundance of Acidobacteria was consistently reduced in the -N treatment compared to the Control in both bulk soils and the rhizosphere (*P* < 0.01), whereas it remained relatively stable in the endosphere (Supplementary Fig. 7). For the -P treatment, most of the phyla in the rhizosphere were specifically reduced relative to the Control (*P* < 0.05), especially the Bacteroidetes, but this was not the case in bulk soils and the endosphere. The bacterial phyla were not strongly impacted by a reduction in K fertilization (-K), with only Saccharibacteria showing a significantly lower rhizosphere abundance than in Control (*P* < 0.01). Meanwhile, the -N treatment significantly increased, and the -P treatment reduced, rhizobial abundance in the rhizosphere in comparison to the Control based on the *rpoB* gene (*P* < 0.05, Supplementary Fig. 8), consistent with the observed increase in the number and diameter of root nodules in the -N treatment relative to the Control (Supplementary Fig. 2). Specifically, *Bradyrhizobium japonicum*, which was the only rhizobial species found in nodules, also

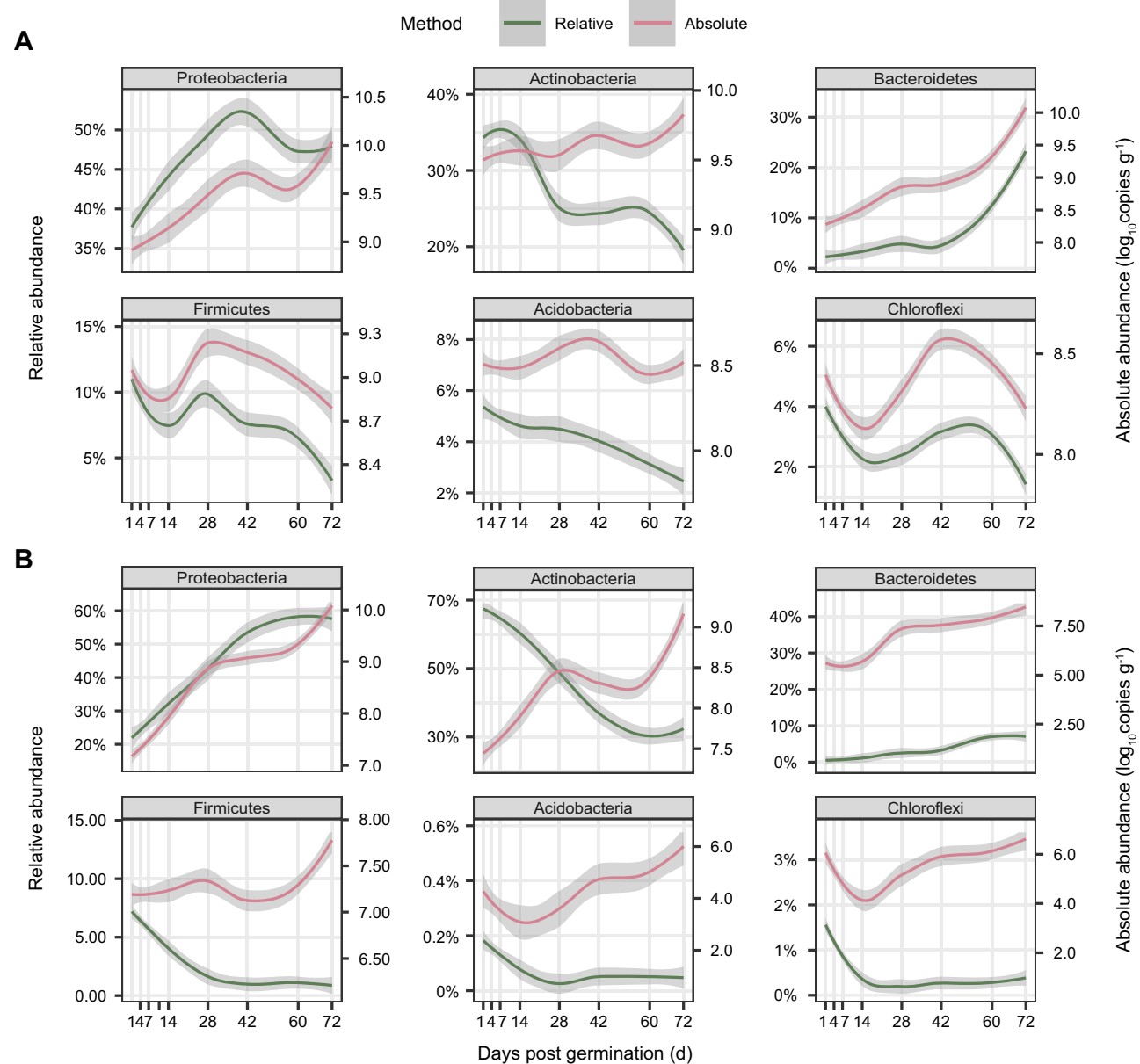

**Fig. 4 | Fitted curves in temporal dynamics of bacterial phyla by QMP and RMP.** Dynamics of bacterial abundance of the six most abundant phyla in the rhizosphere (**A**) and root endosphere (**B**) across plant developmental stages based on QMP (red) and RMP (green) datasets. The Y-axis in the left is the relative abundance (%) of bacterial phyla based on RMP (relative, in green), whereas Y-axis in the right is the absolute abundance ($log_{10}$copies $g^{-1}$) of bacterial phyla based on QMP (absolute, in red). Error bands represent the 95% confidence interval. Source data are provided as a Source Data file.

dominated the rhizosphere, reflecting the exclusive nature of its symbiotic interaction with soybean in the soil (Supplementary Fig. 8).

Considering that the bacterial community in the rhizosphere was more sensitive to fertilization than it was in the endosphere (Fig. 2, Supplementary Fig. 3), we then mainly focused on the taxonomic and functional responses of rhizosphere bacteria to fertilization, and identified core ASVs in the rhizosphere to reduce the complexity of the bacterial community (Fig. 5). We identified 573 ASVs as core taxa, which comprised 2.4% and 74.6% of total bacterial richness and abundance, respectively, that broadly represent the overall bacterial community in each treatment (Fig. 5A, B). Hierarchical clustering analysis illustrated a distinctive pattern of core ASVs between the Control and -N treatment (Fig. 5C). Among the core ASVs, 172 showed a significant difference between the Control and -N treatment, with 69 showing an increase (Fig. 5C, Supplementary Fig. 9). The increased ASVs in the -N treatment were mainly comprised of Rhizobiales (12

ASVs), Sphingomonadales (8), Actinomycetales (8), Rubrobacterales (7) and Gaiellales (6) (Supplementary Fig. 9). For the -P treatment, 67 ASVs displayed significant differences between Control and -P treatment, but most of them showed reduced abundance in the -P treatment, especially the Actinomycetales (24 ASVs), Burkholderiales (7) and Rhizobiales (6). For the -K treatment, however, only 6 ASVs significantly differed from the Control (Supplementary Fig. 9), suggesting a relatively small impact of K fertilization on microbiome composition.

### Functional adaptation of the rhizosphere microbiome

Metagenomic analysis was then used to investigate functional adaptive changes in the rhizosphere microbiome across fertilization treatments (Fig. 6). Detected genes were annotated using KEGG, COG and CAZy databases, and analysis showed a significant reduction of their functional α-diversity in the -N treatment relative to the Control ($P < 0.05$), while the -P and -K treatments were not affected (Fig. 6A–C).

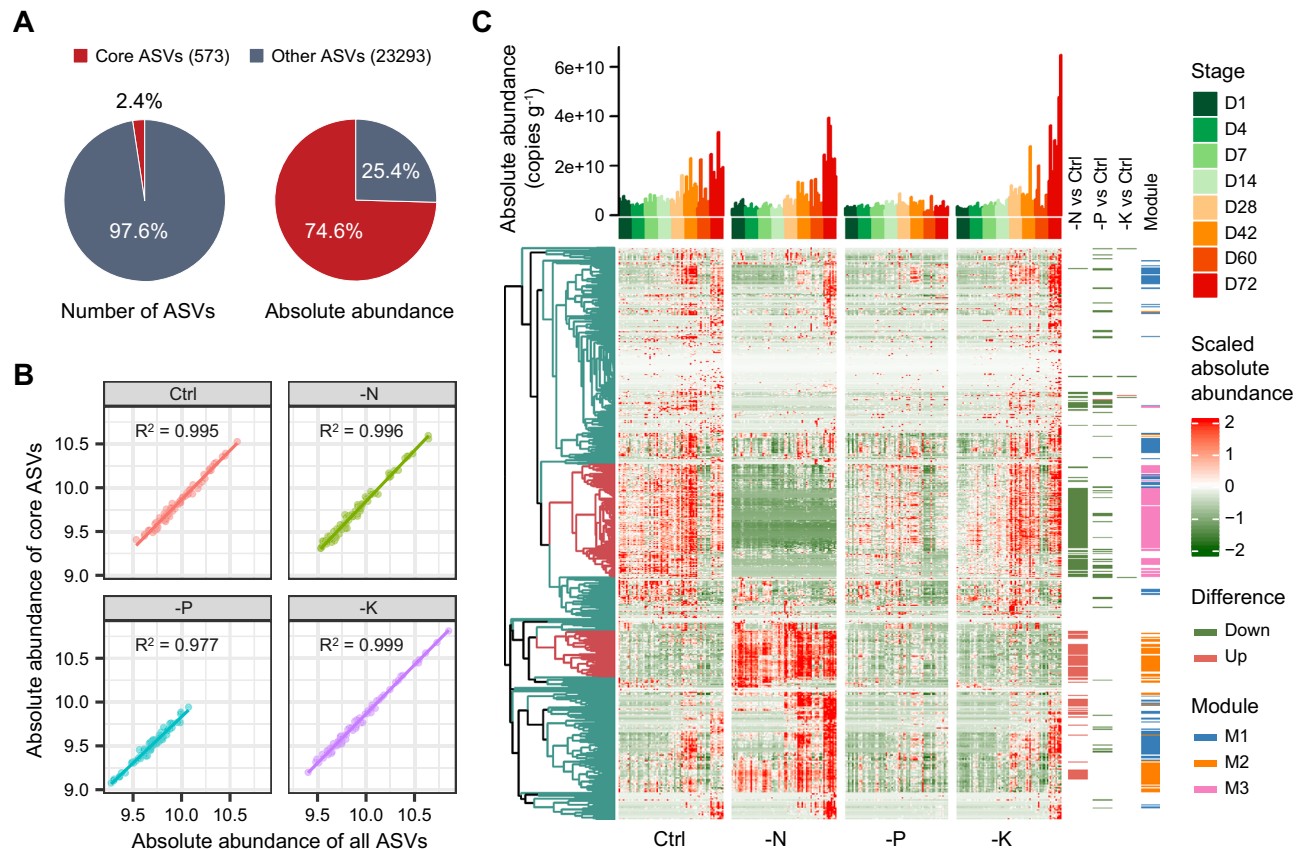

**Fig. 5 | Core ASVs in the rhizosphere. A** richness and accumulative abundance of core ASVs in the rhizosphere. **B** regression coefficient between absolute abundance (log$_{10}$copies g$^{-1}$) of all ASVs and core ASVs in each treatment. **C** clustering heatmap of core ASVs in different treatments and developmental stages. The bar is the accumulative absolute abundance of core ASVs in each sample. The right panel indicates the differentiated ASVs between Control and unbalanced fertilization treatments across plant developmental stages, and the ecological clusters (network modules, in Fig. 7). Source data are provided as a Source Data file.

Constrained principal coordinate analysis (CPCoA) of functional β-diversity showed clear differences among treatments ($P < 0.05$), except for a high overlap between the Control and -K treatment (Fig. 6A–C). We further investigated genes involved in N, P, and K cycling (Fig. 6D). In the -N treatment, genes involved in the N mineralization process, including urea hydrolysis genes (*ureC* and *URE*) and glutamin-(asparagin-)ase genes (*aspQ*), were enriched, whereas those involved in N reduction (e.g., *narB*, *narG* and *narH*), denitrification (e.g., *nirK*), inorganic P solubilization (*gcd* and *ppa*), and K transportation (e.g., *KdpA*, *KtrB*, and *KefB*) processes were specifically depleted, compared with the Control.

For the -P treatment, genes associated with P supply (inorganic P solubilization) and P response (P starvation) were highly enriched (Fig. 6D), and metagenomic assembled genomes (MAGs) further confirmed the presence of genes involved in inorganic P solubilization and P starvation in most MAGs (Supplementary Fig. 10), strongly reflecting conditions of P scarcity. By contrast, genes involved in N mineralization, N reduction, and K transport processes were depleted but nitrification and denitrification processes were enriched in the -P treatment relative to the Control (Fig. 6D). The -K vs. Control comparison revealed a specific enrichment of genes associated with K transport (e.g., *Kup*) in the -K treatment, but only few differences were observed in functional genes involved in nutrient cycling (Fig. 6D).

### Network inference of the core rhizosphere microbiome

Given the significant effect we observed for fertilization regimes on the temporal dynamics of the rhizosphere microbiome, we built co-occurrence networks of core ASVs for different treatments to decipher their co-occurrence patterns across plant development (Supplementary Fig. 11, Supplementary Table 6). Generally, the network complexity (average degree) and robustness gradually increased with plant developmental stages, and the network complexity sharply strengthened at the later stage, implying that the bacterial potential interaction patterns were dynamic and intensive in response to plant development. Next, we compared overall microbial aggregation patterns and identified the keystone hubs in different treatments (Supplementary Fig. 12). We showed that the rhizosphere network complexity was much higher in the -N and -K treatments, with values of 22.48 and 25.12 in the -N and -K treatments, respectively, compared to 7.48 in the Control (Supplementary Fig. 12A). Similarly, the network robustness was significantly higher in all three treatments than in the Control ($P < 0.001$, Supplementary Fig. 12B). For the keystone hubs, we found that there was no module hub identified in each network, except for the presence of a module hub in the network of -P treatment (Supplementary Fig. 12C), suggesting the microbial group rather than a single taxon governed the function of microbial communities in the rhizosphere.

### A low-nitrogen-enriched cluster promotes plant growth

We then constructed a network to compare fertilization-induced changes in ecological clusters (i.e., network modules)[23], and investigated their functions in plant growth promotion using a synthetic community (SynCom) (Fig. 7). Overall, three major modules were identified in the network (Fig. 7A). Module#1 and module#3 were dominated by Proteobacteria, while the taxa in module#2 were mainly composed of Actinobacteria (60%), with Acidobacteria being diminished (Fig. 7B). The bacterial loads of module#2 and module#3 showed opposite responses in the -N treatment relative to the Control, with a significantly higher abundance in module#2 but lower in module#3

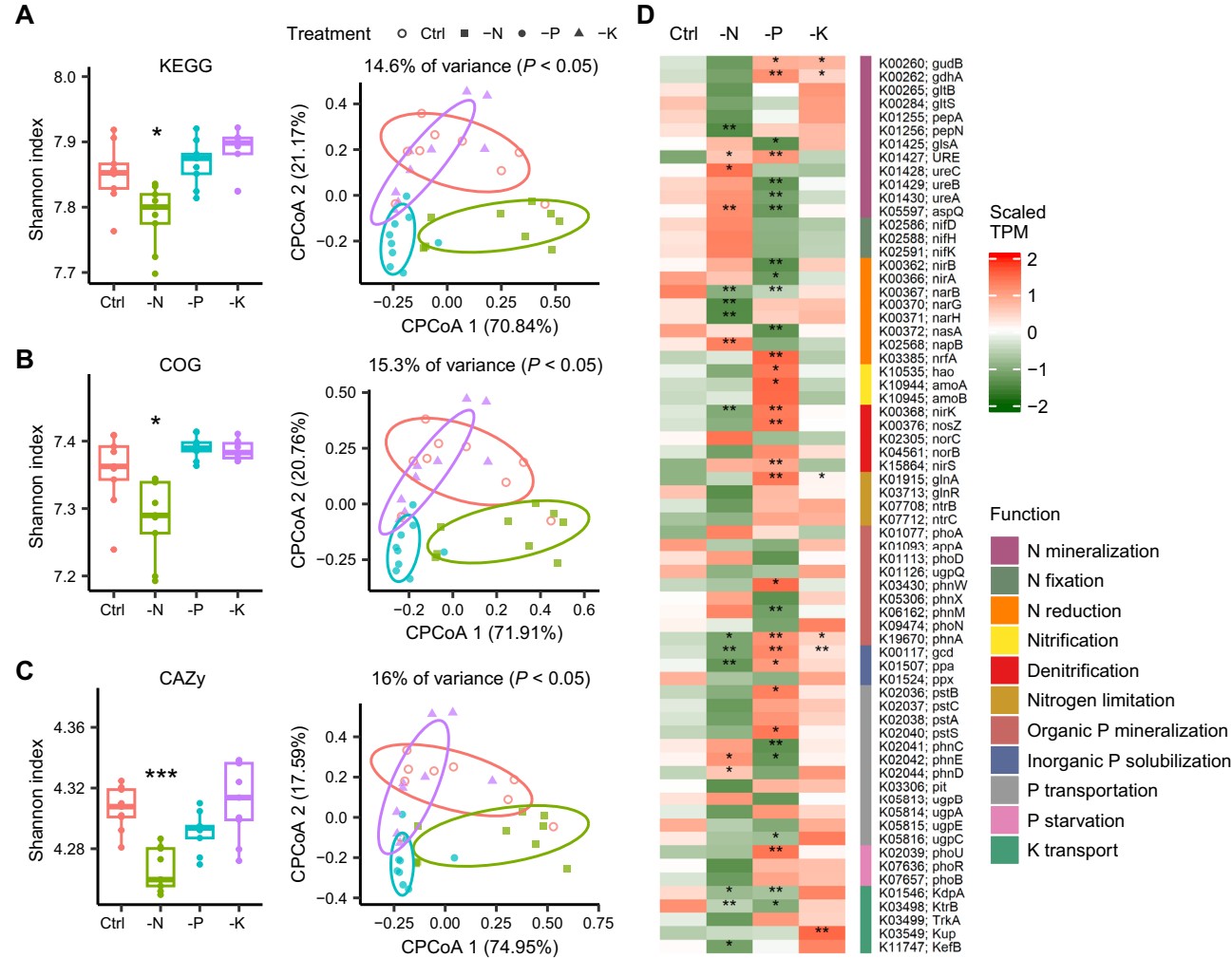

**Fig. 6 | Functional profiles of rhizosphere microbiome in different treatments.** Functional α-diversity (left) and β-diversity (right) of rhizosphere microbiome in each treatment based on the KEGG (**A**), COG (**B**), and CAZy (**C**) databases (n = 9 soil samples). The asterisks represent the level of significance (*P < 0.05, ***P < 0.001) of functional α-diversity between Control and unbalanced fertilization treatments based on Kruskal-Wallis test with Dunn's post hoc analysis. The significance level of functional β-diversity among treatments was assessed by the ANOVA-like

permutation test. Exact P-values are listed in the Source Data file. The box plots indicate the median (center line), the 25th and 75th percentiles (box), and the range of non-outlier values (whiskers). **D** differences of functional genes associated with N, P, and K cycling. The asterisks represent the level of significance (*P < 0.05, **P < 0.01) between Control and unbalanced fertilization treatments based on the paired Wilcoxon test (two-sided). Source data are provided as a Source Data file.

(P < 0.01, Fig. 7C). Intriguingly, the bacterial taxa in module#2 and module#3 exhibited a significant overlap with the increased and decreased ASVs in the -N treatment, respectively (Fig. 5C). For the -K treatment, the abundances of the three modules were comparable with the Control, while for the -P treatment they were consistently reduced (Fig. 7C).

To test the causal relationship between microbial ecological clusters and plant performance, we first isolated 1011 bacterial strains from the soybean rhizosphere by culturing. The isolated bacteria were mainly composed of Proteobacteria (45.0%), Actinobacteria (21.2%), Firmicutes (27.4%) and Bacteroidetes (4.6%), among which 78.9% of strains can be assigned at genus level, with totally 56 genera identified (Supplementary Fig. 13A). After comparing the isolated bacterial strains with core ASVs, we recovered 58% and 49% of the abundances of core ASVs in datasets with accumulative ASV abundances to 60% and 80% of the total sequences, respectively (Supplementary Fig. 13B), indicating a good coverage by isolated strains to rhizosphere core ASVs. In module#2 (i.e., Low-nitrogen-enriched cluster, LNE cluster), seven core ASVs (affiliated with *Rhodococcus*, *Lysobacter*, *Terrabacter*, *Arthrobacter*, *Phyllobacterium*, *Bosea* and *Aeromicrobium*) were

matched with bacterial strains and designated as SynCom7 (Fig. 7D). Considering the bacterial load in module#1 showed non-significant difference between Control and -N treatment, the bacterial strains from module#1 were chosen as a control SynCom (SynCtrl). Overall, 5 ASVs (affiliated with *Brevundimonas*, *Sediminibacterium*, *Mycobacterium*, *Herbaspirillum* and *Sphingomonas*) that matched in module#1 were assembled for use as a SynCtrl (Fig. 7D). To exclude potential Bradyrhizobiaceae-induced nodulation and coordinate the microbial diversity with SynCtrl, five strains derived from LNE cluster were chosen in the experiment (SynCom5) (excluding *Bosea* sp. and *Aeromicrobium* sp. from SynCom7).

Soybeans grown in pots were inoculated with either SynCom5, SynCom7, or SynCtrl in sterilized vermiculite for 2 and 3 weeks. Afterward, the plant phenotypes, including shoot height, total biomass, and nitrate nitrogen concentrations, were measured. The results showed that both SynComs derived from the LNE cluster (SynCom7 and SynCom5) were capable of stimulating plant growth, irrespective of external N addition (Fig. 7E, F, Supplementary Fig. 14). Specifically, the shoot height of soybean inoculated with SynCom5 increased by 15% and 6%, and dry weight increased by 38%

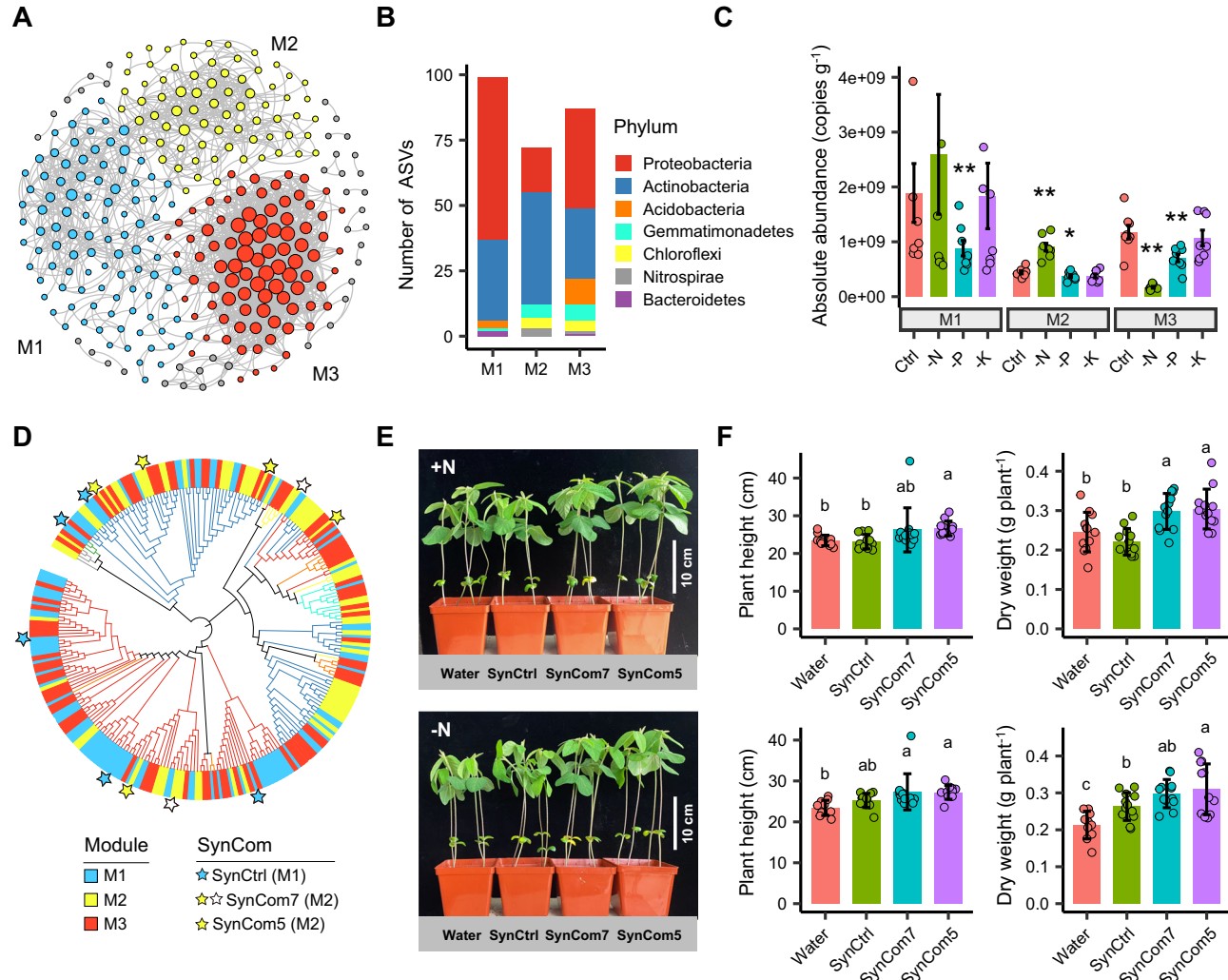

**Fig. 7 | Variance of ecological clusters (network modules) in the rhizosphere and their functions in plant growth promotion. A** visualized network associations and modularity. **B** ASV richness and taxonomy of each module. **C** Cumulative absolute abundance of each module in different treatments (*n* = 8 averaged absolute abundance data for each stage). The asterisks in (**C**) represent the level of significance (**P* < 0.05, ***P* < 0.01) between Control and unbalanced fertilization treatments based on Kruskal-Wallis test with Dunn's post hoc analysis. Exact *P*-values are listed in the Source Data file. Four data points with absolute abundance exceeding 4 × 10⁹ copies g⁻¹ are not shown because of the limitation of Y-axis. **D** Matching modular ASVs with isolated bacterial strains to obtain SynComs.

**E, F** Growth phenotypes of soybeans inoculated with different SynComs at 2 weeks post inoculation under N-amended (+N, upper panel) or N-free (-N, lower panel) conditions (*n* = 10 for -N Water treatment, *n* = 11 for -N SynCom5 treatment, *n* = 12 for other treatments). Different letters indicate significant difference of plant growth phenotypes in different treatments at *P* < 0.05 by one-way ANOVA test with LSD's post hoc analysis (for data fit normal distributions and homogeneous variance) or Kruskal-Wallis test with Dunn's post hoc analysis (for data does not fit normal distributions and homogeneous variance). Exact *P*-values are listed in the Source Data file. The data are presented as mean values ± standard deviation (SD). Source data are provided as a Source Data file.

and 18% at 2 weeks post inoculation, compared with the plant inoculated with SynCtrl with or without N amendment, respectively (Fig. 7F). Similar results were also observed after 3 weeks of inoculation, and the plant nitrate nitrogen contents of SynCom treatments (SynCom7 and SynCom5) increased by 39.5-81.5% and 36.6-51.4% to the SynCtrl treatment in the N-amended and N-free conditions, respectively (Supplementary Fig. 14). Finally, we evaluated the plant growth-promoting (PGP) traits of each SynCom and strain (Supplementary Fig. 15). Compared with SynCtrl, both Syn-Com7 and SynCom5 exhibited higher concentration of indole-3-acetic acid (IAA) and activity of 1-aminocyclopropane-1-carboxylic acid (ACC) deaminase. Two strains in SynCom7 and SynCom5 showed the ability of inorganic phosphorus solubilization. However, the N fixation activity was not detected in all strains, implying the rhizobia might positively interact with SynComs through facilitating plant N nutrition.

## Discussion

Plants are thought to recruit microbial taxa throughout their lifespan, in which time specific root exudates generally drive the microbial colonization pattern in the underground compartments, resulting a highly dynamic nature of root-associated microbiomes[5,21]. Due to the fact that microbial load changes with environment[7–9], RMP fails to detect much of the variance that occurs among samples and does not provide a full picture of the changes that occur over time. Here, for the first time, we reveal the full landscape of changes occurring in soybean root-associated bacteria through application of QMP and suggest that the dynamic root microbiome is able to promote soybean growth. We found evidence that bacterial loads in both the rhizosphere and endosphere exhibited increasing trends with plant development, with most of the major taxa increase steadily over time, despite their inconsistent relative abundance trends (Figs. 3, 4). We observed a stronger succession of the rhizosphere bacterial community across

plant development based on QMP ($0.43–0.54\% \, d^{-1}$) compared with RMP ($0.28–0.34\% \, d^{-1}$), except for the -P treatment (Fig. 2, Supplementary Fig. 4, Supplementary Table 3), suggesting that rhizosphere microbiomes are more dynamic than previous realized. These results, together with an ever-decreasing trend of bacterial α-diversity in the rhizosphere as plant development progresses, suggest that the plants increase microbial carrying capacity but enhance selective pressure as plants develop, which could be largely shaped with long-time plant-microbial co-evolution.

Based on QMP profiling, we found that Proteobacteria, Actinobacteria, and Bacteroidetes were commonly enriched across plant developmental stages (Figs. 3, 4). Bacteroidetes were particularly enriched, followed by Proteobacteria, indicating that they were the most responsive to plant growth (Figs. 3, 4). The Bacteroidetes are commonly recognized as copiotrophic bacteria with r-strategy to flourish in environments with abundant resources[24], with implications for N and P cycling[21,25,26]. The load of Actinobacteria also significantly increased with time, despite their having a reduced relative abundance (Figs. 3, 4), suggesting that plants do not repel Actinobacteria during plant development, but the Actinobacteria are out-grown by Bacteroidetes and Proteobacteria. A pioneer study revealed that Actinobacteria are selectively recruited by active plant cells over inert woody cells[27]. Members of Actinobacteria are known for their ability to produce a range of antimicrobial compounds that promote host fitness[28,29]. Given the fact that plants are more susceptible to disease at early growth stages[30], the temporal complementary of microbial-conferred resistance and plant innate immunity might benefit plant health and development. Together, these results suggest that plant development drives the expansion of the root-associated microbiome, which intimately interacts with plants to enhance host fitness and adaptability.

We found strong evidences that rhizosphere bacteria were more sensitive to fertilization than endosphere (Fig. 2, Supplementary Fig. 3). This could be attributed to the fact that the rhizosphere serves as the interface of plant-soil interactions and is a hotspot for microbial communities compared to the bulk soils, whereas endosphere maintains homeostasis for host health. Particularly, we showed that -P treatment substantially reduced the microbial temporal turnover rate in the rhizosphere, diminished the change of microbial α-diversity, and consistently reduced bacterial load compared to the Control across plant development (Figs. 2, 3), indicating P deprivation delays the development of rhizosphere microbiome. Drought has been reported to similarly delay the maturity of the root microbiome in sorghum and rice[10,11,31], suggesting that the host nutrient supply may be curtailed during periods of severe abiotic stress. Meanwhile, it has been reported that P nutrition has a positive effect on N fixation in soybean[32]. Consistent with this, we observed that soybean root nodules were reduced in number and diameter, and rhizobial abundance in the rhizosphere was decreased under P deprivation (Supplementary Figs. 2, 8). Even so, we observed only a one-fourth of yield reduction in the -P treatment than Control (Fig. 1). For one reason, microbial functional genes encoding inorganic P solubilization and P starvation were specifically enriched in the rhizosphere of -P treatment, and these genes were prevalent in metagenomic assembled genomes (Fig. 6, Supplementary Fig. 10). For another reason, the symbiotic relationship between plants and arbuscular mycorrhizal fungi might be strengthened under low soil P availability[33,34]. We also observed that N transformation processes, including nitrification and denitrification, were significantly enriched in the -P treatment (Fig. 6). These may result from the presence of surplus N in the soil resulting from reduced N uptake by the plants under limiting P, leading to an increase the N transformation processes in the rhizosphere, which corroborates several soil-based studies[35,36].

In contrast, different trajectories were observed between P and N fertilization regimes, with the β-distance of the rhizosphere

microbiome diverging at early stage between the Control and -N treatment, resulting in the parallel development of distinctive rhizosphere microbiome in the two treatments (Fig. 2). This could be attributed to the substantial effect of N on soil chemical and biological processes across various ecosystems[15,37], as well as the legacy effect of N on bacterial communities in the bulk soil over four decades (Fig. 2). The lack of N fertilizer in the -N treatment generally improved soybean root nodulation and increased the abundance of rhizobia (Supplementary Figs. 2, 8), which is consistent with many previous studies showing that a lack of inorganic N inputs and a higher pH benefit symbiotic N fixation[32]. Besides to symbiotic N fixation, N availability can also be improved by organic N mineralization. The increased functional genes involved in N availability but reduced genes related to N reduction in the rhizosphere of -N treatment suggested an effective soybean N supply in absence of N fertilizer (Fig. 6), corresponding with the stabilized soybean productivity in the -N treatment (Fig. 1). Similarly, a large-scale study conducted in the United States and Argentina revealed only an 11% higher soybean yield in the full-N treatment compared to the zero-N treatment, and the positive effect of N fertilizer increased with the increasing yield potential[38], suggesting that the soybean N nutrition can be satisfied across large areas without the need for external N supply.

Root-associated microbiomes have been reported to interact with each other to maintain essential functions in host nutrient acquisition resulting in enhanced plant fitness[39–41]. We found evidence that the co-occurrence patterns of the core bacterial microbiome in the rhizosphere altered in response to fertilization, showing an increased network complexity and stability in the -N and -K treatments relative to the Control (Supplementary Fig. 12), despite there being few taxonomic and functional differences between the Control and -K treatment. It has been reported that a highly connected network can occur under environmental perturbations, such as nutrient scarcity and pathogen invasion[42,43]. An in vitro study examining microbial interactions also suggested that microbes established a high degree of metabolic cooperation under low nutrient availability[44]. The complexity and stability of networks is an important trait for ecosystem function[43], their increase in both -N and -K treatments potentially indicates an increased capacity to alleviate nutrient stress by intensive metabolic interactions such as cross-feeding and facilitation.

The root-associated microbiome is the result of feedback from complex associations between host plants and the soil environment, and extends plant phenotypes[5,45]. However, our understanding of the dynamic changes that occur during these interactions and their effect on plant performance remains elusive. We observed that several ecological clusters (network modules) of core microbial taxa varied in abundance and composition across different treatments (Figs. 5, 7). These taxa may have attributes that confer specialized metabolic functions and affect specific ecosystem processes[41,46–48]. After designing synthetic communities (SynComs) from different ecological clusters, we showed that SynComs from the LNE cluster (SynCom5 and SynCom7) were able to improve soybean growth, irrespective of N addition, but SynComs from module#1 (SynCtrl) had no consistent effect (Fig. 7, Supplementary Fig. 14). We did not observe the formation of root nodules in SynCom treatments, suggesting that the effect was not dependent on the presence of rhizobia. Besides the direct improvement of N availability by rhizosphere microbes, microbes are also known for their PGP effects, such as IAA and ACC deaminase production. The higher production rate of IAA and ACC deaminase in the LNE cluster indicates that plants can benefit from the presence of specific microbial groups through multiple ways (Supplementary Fig. 15). Moreover, it has been reported that the rhizosphere microbiome also plays an important role in facilitating soybean nodulation[49,50], and that nodulation modulates commensal rhizosphere microbiomes[51]. This suggests that positive interactions exist between the rhizosphere microbiome and root nodules. Although our

study focuses on the root-associated bacterial community, it is important to note that other organisms, such as filamentous fungi and protists, may also play a role in shaping network association and ecological functions through "top-down" effects[50,52,53], and how their interactions for benefiting plants through tripartite mutualisms (e.g., rhizobia, mycorrhizal fungi, and plants) requires further investigation[52,53]. Taken together, our study provides experimental evidence for the promotion of specific microbial ecological clusters to increase plant fitness and establishes a framework for identifying, designing and testing agriculturally important microbial SynComs to sustaining crop productivity.

## Methods

### Experimental design, sampling, and measurement of soil chemical properties

The long-term field experiment launched in 1979 was located at Daowai District, Harbin, China (126°51' E, 45°50' N). The climate is typical mid-temperate continental monsoon, and the soil type is classified as Mollisol. Given that the crop can only be cultivated once a year, a 3-year soybean-maize-wheat rotation system was adopted. Four treatments were set, including (1) full-dose fertilization (NPK fertilizer, Control); (2) lack of nitrogen fertilizer (-N); (3) lack of phosphorous fertilizer (-P); and (4) lack of potassium fertilizer (-K) (Fig. 1A). Each treatment consisted of three plots with a completely randomized block design, and each plot comprised 36 m$^2$ (Fig. 1A), and different plots were separated by concrete walls (0.15 m in width and 1.1 m in depth). The nitrogen fertilizer used in the experiment was urea, the phosphorous fertilizer was a combination of $Ca(H_2PO_4)_2 \cdot CaHPO_4$ and $(NH_4)_2HPO_4$, and the potassium fertilizer used was $K_2SO_4$. For soybean cropping, the application rates were 75 kg ha$^{-1}$, 150 kg ha$^{-1}$, and 75 kg ha$^{-1}$ for N, $P_2O_5$, and $K_2O$, respectively, and conventional tillage was applied for soybean planting. Specifically, the ridge was established by a ridging plow, with ridge spacing of 0.65 m. In each ridge, the soybean seeds were sowed in the two rows, with the plant spacing of 0.1 m. Totally, there were around 1000 plants for each plot. The field management, included irrigation and weeding, was adopted according to local farmer practices and was identical for the different treatments. After harvest, the field was fallow until the planting season of next year.

The rhizosphere soil and root samples were collected from the soybean plots in the 2020 planting season. Before sowing, the bulk soils of each treatment were sampled as the background. The soybean variety "Heinong 84" was sown in early May and it germinated after 10 days. We collected the rhizosphere soil and corresponding root samples at 1, 4, 7, 14, 28, 42, 60, and 72 days after germination. The corresponding plant development stages with regard to flowering and pod set are illustrated in Fig. 1A. For each plot, three plants with similar growth state were randomly chosen for collection, and the sampling location was apart from previous sampling stages to prevent the potential marginal effect and disturbance with adjacent plants, with nine biological samples for each treatment at each stage. Briefly, a soybean plant was dug out, and the soil loosely attached to the root was removed by shaking and 5–15 cm of the root was placed in a 50 mL sterilized tube. After sampling, the rhizosphere and root samples were taken to the laboratory in an ice box, and then stored at −80 °C until DNA extraction. At maturity, the yield was measured for each of the soybean plots using standard methods.

The chemical properties of bulk soils were measured to monitor the soil nutrient status. Briefly, soil AHN was measured using alkaline diffusion method. Soil available phosphorus was determined by the molybdenum-blue method after extracting with sodium bicarbonate, and available potassium (AK) was determined by a flame photometer after extracting with ammonium acetate. SOM was determined by chemical oxidation method after digesting with $K_2Cr_2O_7$-$H_2SO_4$. DOC was measured after extraction with 0.5 M $K_2SO_4$ solution. Soil pH was measured with a soil water ratio at 1:5.

### Sample pretreatment and total DNA extraction

The soil remaining tightly adhered to the roots was used as the rhizosphere soil sample. The pretreatment of rhizosphere soil and root samples was performed as previously described in refs. 7,54, with minor modifications. First, 25 mL of PBS solution (containing 137 mM NaCl, 2.7 mM KCl, 10 mM $Na_2HPO_4 \cdot 12H_2O$, 2 mM $KH_2PO_4$) was added to a 50 mL centrifuge tube containing root samples. The mixture was sonicated at 40 Hz for 1 min and shaken to separate rhizosphere soil from the roots. The rhizosphere soil was then transferred to another sterilized 50 mL tube and centrifuged at 9000 rpm for 5 min, after which the precipitated rhizosphere soil was subjected to rotary evaporation (Concentrator Plus, Eppendorf, Germany). Next, the dry rhizosphere soil was homogenized by grinding (Tissuelyser-48, Jingxin, China). After collecting the rhizosphere soil, the remaining roots were sonicated and washed twice in the PBS solution. The nodules were separated from the root using tweezers, and then the roots or nodules were surface sterilized by 5% (w/v) NaClO solution for 1 min, respectively. The surface sterilized roots or nodules were further sonicated and shaken by adding 25 mL of PBS solution and this was repeated three times. Around 200 mg of surface sterilized root or nodules were then subjected to rotary evaporation, and then further homogenized through grinding in liquid nitrogen for extraction of the endophytic microbiome. The bulk soil was also desiccated in the rotary evaporator and then homogenized. The processed samples were then stored at −80 °C.

The total DNA of soil and root/nodule samples was isolated using the FastDNA™ SPIN Kit for Soil (MP Biomedical, Solon, OH, USA) following the manufacturer's protocol. The DNA quality was verified by electrophoresis on a 1% agarose gel. The concentration and purity of the extracted DNA was then confirmed using a NanoDrop 2000 Spectrophotometer (Thermo Fisher Scientific, Inc., USA).

### High-throughput sequencing and bioinformatic analysis

QMP was applied in the high-throughput sequencing of 16S rRNA and *rpoB* genes. This method is used to measure the absolute abundances of total and specific microbial taxa in a sample, through spike-in targeted DNA fragments during PCR amplification[7]. The synthetic spikes (SynSpike) were designed as previously described in refs. 7,8. Overall, 12 sequences of spikes were designed from the V5-V7 region of the 16S rRNA gene, and one sequence was derived from the rhizobial *rpoB* gene. The pUC57 plasmids containing each spike sequence were transferred into *Escherichia coli* TOP10 and then extracted using a TIANpure Midi Plasmid Kit (Tiangen Biotec, China) (Supplementary Fig. 16). Different concentrations of spikes were combined to form the SynSpike for sequencing of soil, root and nodule samples. Finally, 30 μL of the extracted DNA was mixed with 10 μL of the SynSpike for PCR amplification. The V5-V7 region of bacterial 16S rRNA gene, and *rpoB* gene of symbiotic rhizobia were amplified through primer pairs 799F (5'-AAC MGG ATT AGA TAC CCK G-3') and 1193 R (5'-ACG TCA TCC CCA CCT TCC-3')[55], and rpoB1479F (5'-GAT CGA RAC GCC GGA AGG-3') and rpoB1831R (5'-TGC ATG TTC GAR CCC AT-3')[7], respectively. The resultant 16S rRNA and *rpoB* derived PCR products were sequenced on the MGISEQ-2000 platform of BGISEQ using the single-end 400 bp (SE400) module in BGI Shenzhen (BGI, Shenzhen, China).

The microbial bioinformatic analysis was performed with QIIME 2 2021.11[56]. The raw sequencing data was demultiplexed and filtered using the q2-demux plugin followed by denoising with DADA2[57]. The amplicon sequence variants (ASVs) of 16S rRNA and *rpoB* genes were annotated against the Greengenes[58] and the rhizobial *rpoB* reference sequence databases[7], using classify-sklearn naive bayes taxonomy classifier and classify-consensus-blast taxonomy classifier, respectively. For 16S rRNA sequencing data, the non-bacterial ASVs in all samples and Bradyrhizobiaceae-affiliated ASVs in the root endosphere were filtered out, as the abundant Bradyrhizobiaceae from the root nodule symbiosis might adversely affect the estimation of bacterial

diversity in the endosphere. For the *rpoB* sequencing data, only ASVs annotated at the species level were retained. The filtered ASVs were aligned with MAFFT[59] to construct a phylogenetic tree using FastTree2[60]. Bacterial α-diversity metrics, β-diversity metrics, and principal coordinate analyses (PCoA) were estimated using q2-diversity plugin after rarefaction (1030 sequences per sample) for RMP, and using the R package "vegan" without rarefaction for QMP. The rarefaction curve showed that sequencing depth was suitable to calculate Shannon index of bacterial communities (Supplementary Fig. 17).

## Metagenomic sequencing, assembly, and binning

The rhizosphere samples from days 1, 42, and 72 after germination were subjected to metagenomic sequencing on the DNBSEQ-T7 platform of BGISEQ using the paired-end 100 bp (PE100) module in BGI Shenzhen (BGI, Shenzhen, China). The raw sequencing data was assembled using the Easy Metagenome analysis procedure[61]. Briefly, raw reads were filtered to remove low-quality and host (*Glycine max*) genomic sequences using KneadData (v0.10.0). Metagenome assembly of individual samples was processed by MEGAHIT (v1.2.9) using the default parameters[62]. Bowtie 2 (v2.4.4) was used to map the raw reads to assembled contigs to calculate the read coverage of each contig, after evaluating the quality of contigs by QUAST (v5.0.2)[63,64]. The contigs were subjected open reading frame (ORF) prediction using Prodigal (v2.6.3)[65]. Afterward, a non-redundant gene catalog was constructed using CD-HIT (v4.8.1)[66]. The raw reads and normalized TPM (Transcripts Per kilobase Million) data was quantified after BLAST against the non-redundant gene catalog by Salmon (v1.6.0)[67]. The non-redundant genes were annotated using KEGG and COG databases by eggNOG-mapper (v2.1.6)[68], and using CAZy database by DIAMOND[69,70].

For metagenomic binning, contigs longer than 2000 bp were grouped into 6153 bins using MaxBin2[71] and MetaBAT2[72] with the default parameters. CheckM (v.1.0.12)[73] was used to assess the assembly quality of all bins. Totally, 655 bins with completeness ≥ 70% and contamination ≤ 10% were obtained. All filtered bins were aggregated and dereplicated using dRep (v.3.2.2)[74] with the default overlap threshold (-nc 0.1), and were clustered into strain level and species-level genome bins at 99% and 95% of the average nucleotide identity, respectively. Finally, we obtained a non-redundant set of 171 bins (MAGs) at the strain level and 140 MAGs at the species level.

## Co-occurrence network analysis

The core bacterial ASVs (with relative abundance > 0.1% for at least one sample) in the rhizosphere were selected for network construction. For each treatment, we constructed three networks to infer the dynamic co-occurrence patterns of core ASVs with plant development by separating samples into three stages (D1-D14, D28-D42, and D60-D72), and built an aggregation of microbial association network to compare fertilization-induced change on microbial co-occurrence patterns and identify the keystone hubs. We also combined all rhizosphere samples to find microbial network modules of core bacterial ASVs. A pairwise Spearman correlation matrix was calculated based on the absolute abundance of the core ASVs with the "corr. test" function in the package "psych" in R (v4.1.0). The *P* values were adjusted with the false discovery rate (FDR) method. A cutoff for the Spearman's rank correlation coefficient (ρ) of higher than 0.8 or lower than −0.8 with $P < 0.05$ was used for network construction. Network properties were characterized via the "igraph" package in R, and network robustness was calculated as previously described in ref. 75. The modules in a network, defined as sub-communities of highly interconnected nodes, are recognized as microbial ecological clusters performing different functions[23]. The identification of ecological clusters in the network was performed in Gephi (v0.9.5) platform, using default parameters[76]. Putative keystone hubs in the network were identified according to its within-module connectivity ($Zi$) and

among-module connectivity ($Pi$). Nodes with $Zi > 2.5$ were identified as modules hubs, whereas nodes with $Pi > 0.62$ were identified as connectors[75,77]. All networks were visualized in Gephi (v0.9.5)[76].

## Rhizosphere microbial isolation and experimental validation

The rhizosphere bacterial strains were isolated using LB and R2B mediums. Briefly, 3 g of rhizosphere soil from each treatment was placed in a sterile centrifuge tube containing 30 mL of PBS buffer, and then 10× serial diluted using PBS buffer. 50 μL of each of the $10^{-4}$, $10^{-5}$, and $10^{-6}$ dilutions were placed on the surface of the medium. For the isolation of spore-forming bacteria, 3 g of rhizosphere soil was stored in a desiccant for 2 weeks and heated at 80 °C for 30 min after mixing with 30 mL PBS buffer, followed by dilution and plating as described above. All culture dishes were incubated at 30 °C for 48 h, and the bacterial colonies on the plate were picked for further purification. The universal primer pair 27 F/1541 R was used for bacterial identification. The purified bacterial strains were then stored in 30% glycerol solution at −80 °C.

The nodes (ASVs) in the co-occurrence network of each module from Fig. 7 were BLASTed against the isolated bacterial strains, and strains with > 95% identity for the 16S rRNA gene were considered to belong to the same ASVs. For the ASVs affiliating to more strains, we chose the strain with the highest similarity to improve the representativeness. Afterwards, we selected one strain for each genus to reduce the complexity of synthetic communities. Considering only -N treatment in the module#2 showed significantly higher bacterial loads than that in Control, we chose bacterial strains in module#2 as synthetic communities in the validation experiment, and took bacterial strains in module#1 as the control SynCom (no significant change of bacterial loads between two treatments). Overall, 7 strains (i.e., *Rhodococcus* sp., *Lysobacter* sp., *Terrabacter* sp., *Arthrobacter* sp., *Phyllobacterium* sp., *Bosea* sp. and *Aeromicrobium* sp.) in the Module#2 were designated as SynCom, and five strains in the module#1 were designated as SynCtrl (i.e., *Brevundimonas* sp., *Sediminibacterium* sp., *Mycobacterium* sp., *Herbaspirillum* sp. and *Sphingomonas* sp.). To exclude potential Bradyrhizobiaceae-induced nodulation and coordinate the microbial diversity with SynCtrl, 5 strains from SynCom7 were further selected (excluding *Bosea* sp. and *Aeromicrobium* sp. from SynCom7), which was designated as SynCom5.

We then tested the effect of these SynComs on soybean plant growth. To do this we used the following treatments: Control (sterilized water), SynCtrl (5 strains from Module#1), SynCom7 (7 strains from Module#2), and SynCom5 (5 strains from Module#2). Sterilized vermiculite was used as substrate and MFP mediums with or without nitrogen (i.e., $NH_4NO_3$) were applied as nutrient solutions. The soybean seeds were surface sterilized with 5% (w/v) NaClO for 5 min and then planted in sterilized vermiculite. Each treatment had 12 seedlings. The SynComs were prepared by centrifuging the bacteria solution at 9000 rpm for 2 min and re-suspended in sterilized distilled water adjusted to a final optical density (OD) of 0.1 for each strain. After seed germination (4 days after planting), 1 mL of the SynComs or sterilized water was inoculated into the roots, and this was repeated after 1 week. The shoot height and total biomass of the soybean plants were measured 2 and 3 weeks after first inoculation, and the nitrate nitrogen concentrations of whole soybean plants were measured using samples taken from 3 weeks after first inoculation. The concentration of plant nitrate nitrogen was measured by the colorimetric method after digestion with salicylic acid-$H_2SO_4$ and NaOH solution.

## Characterization of plant growth-promoting (PGP) traits in synthetic communities

The PGP functions of each bacterial strain and synthetic community, including IAA production, ACC deaminase activity, inorganic phosphorus solubilization and nitrogen fixation, were investigated in vitro. IAA concentrations were determined as described in ref. 78, with some modifications. Briefly, the Landy medium was used to culture strains

for 3 days and 200 rpm at 28 °C. Afterwards, 1 mL of the supernatant (obtained by centrifugation at 9000 rpm for 5 min) was mixed with 2 mL of Salkowski's reagent (150 mL of concentrated $H_2SO_4$, 250 mL of distilled $H_2O$, 7.5 mL of 0.5 M $FeCl_3·6H_2O$) and incubated in darkness for 20 min. The IAA concentration was determined by colorimetric method at 530 nm, and a standard curve was prepared by a series of pure IAA concentrations. To test the ACC deaminase activity of synthetic communities, bacterial strains were inoculated in DF salts medium supplemented with ACC as the sole nitrogen source for 3 days and 200 rpm at 28 °C. ACC deaminase activity was quantified using the method as previously described in ref. 79. The amount of α-ketobutyrate produced by each SynCom was determined by a standard curve of pure α-ketobutyrate at 540 nm. Inorganic phosphate solubilizing capacity of the bacterial strains was tested on the Pikovskaya's agar medium as described in ref. 80. Halo zones surrounding the colonies were measured after 7 days of culture at 28 °C. Nitrogenase activity of the bacterial strains was measured by acetylene reduction assay as described in ref. 81. Briefly, bacterial strains were cultured in the modified Döbereiner medium for 3 days and 200 rpm at 28 °C. Then, the bacterial culture was transferred to the serum vial and 10% of headspace gas was replaced with acetylene. The ethylene concentration was determined with a gas chromatograph (HP7890B, Agilent, USA) after inoculation for 24 h at 28 °C.

## Statistical analysis

All statistical analyses were performed in R (v4.1.0, http://www.r-project.org). The data for soil chemical properties and soybean phenotypes were tested for normality using the Shapiro-Wilk test and homogeneity of variance using Bartlett's test. For data with normal distributions and homogeneous variance, one-way ANOVA test with Dunnett's or LSD's post hoc analyses was used for multiple comparisons. Kruskal-Wallis tests with Dunn's post hoc analysis for multiple comparisons were applied to data with non-Gaussian distributions or heteroscedasticity.

The linear mixed model analysis was performed to identify the major drivers of microbial α-diversity and abundance using the R package "lme4"[82]. Nonparametric statistical tests (Kruskal-Wallis tests with Dunn's post hoc analysis) were performed to evaluate the difference of α-diversity between different development stages and different fertilization treatments. The relative contribution of different factors on bacterial Bray-Curtis dissimilarity was tested with PERMANOVA using the Adonis2 function in R package "vegan"[83]. Linear regression analysis was used to examine the relationship between temporal distance and Bray-Curtis distance between two samples in each treatment and between sampling stage and Bray-Curtis distance of each unbalanced fertilization treatment to the Control. Paired Wilcoxon test was applied to test the significance of bacterial phyla, core ASVs, and functional genes between Control and each unbalanced fertilization treatment, after averaging the abundance of bacterial taxa in each stage. The significantly changed ASVs to unbalanced fertilization were defined as the $P < 0.05$ by paired Wilcoxon test and fold change > 1.5. All statistical test $P$-values were corrected using the FDR method, except for those obtained with the paired Wilcoxon test. For all statistical tests, $P < 0.05$ were considered significant, and individual $P$-values are specified in figure legends. Data visualizations were primarily generated using the "ggplot2" R package. Heatmaps were generated using the "ComplexHeatmap" R package[84]. The phylogenetic tree was primarily constructed by 16S rRNA gene, except that the phylogenetic tree of MAGs was constructed based on the complete genomes. The visualization of phylogenetic tree and data annotation were conducted with iTOL v6[85].

## Reporting summary

Further information on research design is available in the Nature Portfolio Reporting Summary linked to this article.

## Data availability

The 16S rRNA amplicon data (SAMN32647504-SAMN32648110), *rpoB* amplicon data (SAMN32650334-SAMN32650836), and metagenome data (SAMN32727371-SAMN32727406) associated with this study have been deposited in the NCBI Sequence Read Archive (SRA) under the project accession PRJNA922226. The classification information of the metagenomic assembled genomes (MAGs) and isolated bacterial strains has been uploaded to the GitHub repository: https://github.com/mxwang2016/Soybean. Source data are provided with this paper.

## Code availability

The R scripts for calculating and visualizing microbial diversity, load, and composition have been uploaded to the GitHub repository: https://github.com/mxwang2016/Soybean.

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

## Acknowledgements

We would like to thank Prof. Li Luo and Mr. Ningning Li for their generous help in bacterial strain isolation and identification. E.W. was supported by CAS Project for Young Scientists in Basic Research (YSBR-011), National Key R&D Program of China (2022YFF1001800), the National Science Foundation (32088102, 32050081, 31825003, 32001886 and 31870218), the Strategic Priority Research Program "Molecular Mechanism of Plant Growth and Development" of the Chinese Academy of Sciences (XDB27040207), and the New Cornerstone Science Foundation through the New Cornerstone Investigator Program and the XPLORER PRIZE. X.C. was supported by the Strategic Priority Research Program of the Chinese Academy of Sciences (XDA28030103).

## Author contributions

E.W., M.W., X.M., and X.W. conceived and designed this research; M.W., X.M., A.-H.G., X.W., Q.X., L.W., X.S., M.J., Y.W., and H.L. performed experiments; M.W., A.-H.G., and X.W. analyzed data; A.-H.G., M.W., E.W., J.D.M., and W.Y. wrote the paper; E.W. and X.C. oversaw the entire study. All authors read and approved the manuscript.

## Competing interests

The authors declare no competing interests.
