## [Peer Review File · Nature Communications]

REVIEWER COMMENTS

Reviewer #1 (Remarks to the Author):

This study investigated the temporal dynamics of the soybean root-associated microbiome using quantitative microbiome profiling (QMP). The study aimed to understand the response of the microbiome to imbalanced fertilizer treatments (i.e., N, P, or K deficiency) and validates the beneficial effects of low nitrogen-enriched microbes on plant growth using synthetic microbial communities.

The manuscript is well-written and provides detailed and logical explanations. I have a few suggestions for improvement:

Does the network analysis in Fig 6 used to obtain the three SynComs include samples from all nutrient-deficient treatments? Why not analysing the core species from nitrogen-deficient treatment network (Fig 5) as members of the low-nitrogen-enriched SynCom?

The network analysis is based on correlation inference rather than actual microbial interactions. Please elaborate on: What are the nitrogen fixation efficiencies or other beneficial traits related to nitrogen cycling of the three SynComs, especially the SynCom composed of theoretical core species enriched under low nitrogen conditions? Do they exhibit any differences in nitrogen fixation efficiency, for example? Besides, the bacterial loads in module 2 was significantly different in both -N and -P treatments compared with control which might make the selection basis for SynComs confusing. I believe it would be helpful to provide some validations on their functional traits to build convincing connections between sequencing data and cultivation-dependent method.

Please provide the classification information for these MAGs. These MAGs are derived from different samples. Could you please provide some data to support their distribution across different treatments? Do they belong to abundant groups or rare groups? For nitrogen cycling, genes related to N fixation process were not found in these genomes, what are the possible reasons? Was the phylogenetic tree in Supp Fig 12 constructed based on 16S sequences annotated from complete genomes? To what extent could these MAGs cover the isolated cultures?

L145-146: The statements seem to contradict the results in Fig 2D. In Fig 2, the potassium-deficient treatment but not the nitrogen-deficient treatment grouped separately.

L273: Have these bacteria been identified at the species level? How many unique species were included? I believe it would be better to provide basic classification information for these 1011 bacterial strains. How many strains performed >95% for the 16S rRNA gene with the same core ASVs? As I considered,

each ASV can match multiple isolates. Did you randomly select one strain from the matched results as the SynCom member? Or what was the basis for this selection here?

In Fig 3A, does the term "relative abundance" refer to relative data derived from QMP? Or data from RMP?

In Fig 6, does "feature number" refer to the number of ASVs?

Reviewer #2 (Remarks to the Author):

The manuscript by Wang and colleagues is an interesting one, rich in results. This richness, however, can be of risk while sometime the authors wish to provide with as many results as possible without presenting/discussing them properly.

Some of the strong points of the analysis were considered as supplementary. For instance, the methodological comparison between QMP and RMP should be better presented at the top of Results section. The same applies to the functional (metagenomics) analysis which now is completely.

Another element of confusion is the fact that some of the analyses were carried out on both endosphere and rhizosphere, whereas others just on rhizosphere (i.e., network analysis). Results are sometime only presented for a compartment, like the functional analysis only reported for rhizosphere in Results while being done also on endosphere (see line 452 in Methods). explanations on the logic for these swings would be needed.

While the part on SynCom is interesting I see this more as later add-on the authors gave included to further enrich manuscript. Indeed, methodology is quickly described and often difficult to follow. Also, the analysis on the plant growth was limited to just two measured parameters (shoot height and total biomass) at 2 weeks after germination. Authors may consider to not include this part and present in a separate, more detailed, publication. If they want to have it, then methodology for constructing the network to compare fertilization-induced changes in ecological clusters has to be better explained in both Results and Methods.

Finally, I found the Discussion poor compared to the amount of findings. An assessment of the differences found and the possible reasons behind between bulk soil, endosphere and rhizosphere in all

the treatments is missing. Also, an evaluation of the advantages of QMP over RMP is lacking. Same for the functional (metagenomics) findings.

Other comments:

- it would be important to add hypothesis at the bottom of the Introduction. This may also help in figuring out what are key results to be presented and, thus, better structuring the following sections.

- lines 134-137: how you can say that that time corresponds to the onset of root nodulation? Please provide evidence/reference for that.

- lines 145-146: in fig 2D it seems -K not -N treatment that separates from others.

- how were the DNA-read samples normalised? What methodology (e.g., SRS) was applied?

- p-values are missing in all figures

- Figure 2A: were samples significantly different? Add letters to show this.

- Figures 2G-H are very difficult to read, please find a better way to present them. Also, on the x-axis what is the difference between "temporal distance" and "days of germination"?

- Figure 3: are the significant differences among temporal stages for all the 4 treatments? Also, in the phyla barplots it is impossible to distinguish between treatments.

- Figure 3 refers to 16S data? and Supp Fig. 9 to rpoB? Please clearly indicate this in the legend.

- Lines 335-337: not clear, to be rephrased.

- Lines 495-508: how modules were generated? Please add a paragraph on top to explain the method. As it is, it is very hard to be understood.

- in Results I would call it "soil" not "Mollisol".

Reviewer #3 (Remarks to the Author):

Comment to Wang et al., Nature Communication NCOMMS-23-22973

Title: Dynamic root microbiome sustains soybean productivity over four-decades of unbalanced fertilization. The manuscript under consideration investigates temporal dynamics (days) of the root-associated bacterial community of soybean plants of a field trial with fertilization treatments (-N, -P, -K). The authors use metagenomics sequencing (16S rRNA and rpoB) and Co-occurrence network analysis to investigate the changes in specific bacterial groups across plant growth stages. In the last part, the authors selected and isolated three microbial communities from the field to test the plant growth-promoting functions in a pot experimental setup.

This is an extensive study presenting very detailed information on changes in microbial (bacterial communities) over time and during the growing season. The presentation is relatively clear and asks relevant and novel research questions. The experimental design is elegant, and the complex metagenomics results are well- presented and analyzed. However, the paper lacks suitable hypotheses and conclusions. The fact that the experiment was analyzed after a few decades of fertilization treatments needs more attention and analysis. In addition, more temporal dynamic analyses of the Co-occurrence bacterial network can improve the manuscript's conclusions and key messages. Finally, the verification experiment done by the authors is crucial and powerful. However, the results of specific communities of plant-promoting function need more connection to the nutrient status and specifically nitrogen status. The other nutrient limitation needs to be presented and experimented with older plants.

Below I mentioned a few aspects that can be improved in the manuscript to provide a more robust and holistic conclusion on the temporal resolution of plant-soil interaction.

Major comments-

1. Long-term effect of fertilization – The four-decade effect of unbalanced fertilization needs proper attention, analysis, and discussion. Only one figure (fig. 1C) presents the soybean yield differences in 2017 and 2020; no long-term effect is presented. The slight reduction in yield is shown only in the –P treatment and not in the –N –K. This no reduction in yield raises questions about the relevance of the fertilization treatments. If the data is unavailable, the title and the abstract should be modified accordingly.
2. Co-occurrence network temporal resolution – The temporal resolution of the Metagenomics sequencing is nicely presented, leading to interesting community structure differences among the fertilization treatments. However, in the co-occurrence network analysis, only an aggregation of the temporal resolution is analyzed (according to my understanding). I suggest dividing the time point into several stages (for example, early growth, mid-growth, and late) and analyzing the Co-occurrence network separately. Observing the temporal resolution of the bacterial community (Fig. 2), It seems that the most dominant changes are in D14 and D42, and D60 (Shannon index Fig. 2C). In addition, analyzing the temporal co-occurrence network can provide more specific bacterial community to provide more robust results in the experimental validation stage.
3. Keystone taxa – The idea of keystone taxa has become an important and encouraging research topic. Identifying a specific species of bacteria as keystone taxa can improve the paper's conclusions. See recent publications (Amit and Bashan "Top-down identification of keystone taxa in the microbiome" Nat

Comm 2023; Banerjee et al., Keystone taxa as drivers of microbiome structure and functioning Nature Reviews 2018).

4. Discussion about other organisms – The manuscript analyzes and discusses only the interaction between plants and bacteria. However, the effect of other organisms, such as fungi is not presented and this a weakness. Fungi, specifically arbuscular mycorrhiza, have been shown to influence the microbial community and the co-occurrence of the microbial network. Specifically, discussing the –P treatment requires the potential influent of nutrient supply by the fungi. Moreover, discussion about N and P limitations and the tripartite mutualisms (rhizobium:mycorrhizal:plant) can be discussed too (See Deveau et al., Bacterial–fungal interactions: ecology, mechanisms and challenges FEMS Microbiology 2018).

5. Validation experiment – Plants were measured two weeks after germination. More extended experiments (several weeks) can produce more robust results and effects on plant growth. In addition, a fertilization treatment to validate the exact microbial community to nutrient availability would be useful.

6. Title: the title does not really reflect the content (e.g. Dynamic root microbiome sustains soybean productivity over four-decades of unbalanced fertilization). The work is mainly on describing microbial communities in relation to fertilisation treatment and time after planting.

7) A range of papers already showed microbiome dynamics in plants (or plant roots). The novelty of this work compared to other studies is unclear.

Minor comments

Abstract

The word bacteria should be mentioned. Microbiome includes many organisms, however, the study is focused only on bacteria. In addition, the manuscript's conclusions mentioned in the abstract can be more focused and less general.

What is meant with quantitative microbiome profiling (this is not really discussed in the methods, as far as I can see, but possibly I overlooked this).

Introduction

Line 73: Starting the paragraph with “specifically” is strange

Line 85: The term “microbial priming” should be mentioned (Bastida et al., Global ecological predictors of the soil priming effect” Nat Comm 2019; Oppenheimer-Shaanan “A dynamic rhizosphere interplay between tree roots and soil bacteria under drought stress” elife 2023).

Line 97: General hypotheses are missing.

Methods

Line 390: More information about soil management is important such as irrigation, tillage, and cover crop.

The sampling method requires more details. What is the percentage of sampled plants per plot? How does the number of sampled plants affect competition between the plants?

Line 415: What instrument was used to grind the samples?

Line 450: The rarefaction curves can be useful in the supplementary information.

Line 490: Why a different set of primers was used?

Results

Fig. 2: The X-axis in panels 1B and 1C are misleading; the number of days between the time points is not equal (4 days, 3 days, 7 days, 17 days...). Changes happening at the beginning of the growth stage (days 1-7) are much quicker than the later stages (days 28-42).

Line 121: This statement requires more experiments. The nitrogen amount is significantly less depleted compared to the other nutrients.

Figure 3: Why the -N treatment has such a high standard error (BS).

Line 174: The significant effect is in the rhizosphere and less in the endosphere. Analysis comparing the two can be important.

Fig S7: The figure is very nice and important. I will consider moving it as a main figure instead of Fig. 3. Please mention the different y-axis units in the different panels in the text.

Line 190: Day 14 is a critical day with many changes. Why? What happened after two weeks? This may be relevant to the rhizobia establishment process.

Line 190: I do not get the last point. Why the Bacteroidetes group is one of the most responsive to plant growth? Maybe to plant growth stages. The Proteobacteria show the same pattern.

Fig S8: At which time point is this date based?

Line 255: Belong to the discussion, and the results don't support this claim.

Fig. S11: very nice figure. I will move the figure to the main part of the article.

Discussion

Line 315: "Bacteroidetes were particularly enriched, indicating that they were the most responsive to plant growth". Proteobacteria show the same pattern.

Line 337-341: All these results came from the same experiment? If yes, add in which exact figure.

Line 329-348- The paragraph needs to be organized better. I assume you tried to discuss the effect of environmental parameters on the microbial community, but the structure needs to be written better.

Line 369: How and where was it shown that the module#1 effect is inconsistent and probably not rhizobia dependent effect?

REVIEWER COMMENTS

Reviewer #1 (Remarks to the Author):

This study investigated the temporal dynamics of the soybean root-associated microbiome using quantitative microbiome profiling (QMP). The study aimed to understand the response of the microbiome to imbalanced fertilizer treatments (i.e., N, P, or K deficiency) and validates the beneficial effects of low nitrogen-enriched microbes on plant growth using synthetic microbial communities.

The manuscript is well-written and provides detailed and logical explanations. I have a few suggestions for improvement:

Does the network analysis in Fig 6 used to obtain the three SynComs include samples from all nutrient-deficient treatments? Why not analysing the core species from nitrogen-deficient treatment network (Fig 5) as members of the low-nitrogen-enriched SynCom?

Response: We appreciate for your positive comments and suggestions. Yes, the network in Fig. 6 (Fig. 7 in the revision) is constructed through combining all samples from the rhizosphere in all four treatments to display the overall microbial co-occurrence pattern. In the Fig. 5 (Supplementary Fig. 12 in the revision), we intended to find the differences in network topologies across four treatments, but we found that there was no keystone species identified in the -N treatment (module hubs based on Z_i - P_i scores, see figure below), which could be attributed to the microbial groups rather a single taxon governing the function of microbial communities in the rhizosphere of -N treatment. Therefore, we provided an additional network analysis in Fig. 6 (Fig. 7 in the revision) to compare the change of abundance in ecological clusters (network modules) and found a low-nitrogen-enriched ecological cluster (Module#2), in which most of taxa were significantly enriched in the -N treatment compared with the Control (Fig. 5C in the revision). We have provided more detailed information towards the logic flow between Supplementary Fig. 12 and Fig. 7 in the revision, please see L263-266 in the revised manuscript. We further displayed the comparison among core species, low-nitrogen-enriched species and ecological cluster for clearer data presentation in the revised Fig. 5C, following your suggestions.

Supplementary Fig. 12C Identified keystone taxa of co-occurrence networks in the rhizosphere of each treatment.

The network analysis is based on correlation inference rather than actual microbial interactions. Please elaborate on: What are the nitrogen fixation efficiencies or other beneficial traits related to nitrogen cycling of the three SynComs, especially the SynCom composed of theoretical core species enriched under low nitrogen conditions? Do they exhibit any differences in nitrogen fixation efficiency, for example? Besides, the bacterial loads in module 2 was significantly different in both -N and -P treatments compared with control which might make the selection basis for SynComs confusing. I believe it would be helpful to provide some validations on their functional traits to

build convincing connections between sequencing data and cultivation-dependent method.

Response: Thank you for your suggestions. We have conducted more experiments in revealing the plant growth-promoting traits of SynComs (Supplementary Fig. 15 in the revision). Surprisingly, the bacterial strains in these SynComs showed no nitrogen-fixing efficiency, but some strains exhibited abilities in IAA and ACC deaminase production, as well as phosphorous solubilization. In particular, both SynCom7 and SynCom5 showed significantly higher activities in producing IAA and ACC deaminase than that in SynCtrl, and phosphorous solubilization was exclusively observed in SynCom7 and SynCom5. These results suggest that SynCom7 and SynCom5 endow plant benefits through multiple mechanisms, and the rhizobia may positively interact with SynComs for nitrogen nutrition. Although both -N and -P treatments significantly changed the bacterial loads in Module#2 compared with Control, they showed opposite trends, in which bacterial loads in Module#2 enriched in the -N treatment, but reduced in -P treatment. Therefore, we chose bacterial strains in Module#2 as SynComs for experimental validation. We have provided the functional traits of SynComs in Supplementary Fig. 15 in the revised manuscript, and made corresponding revisions in the Results part (L307-313).

Supplementary Fig. 15 Plant growth-promoting (PGP) functions of synthetic bacterial communities.

Please provide the classification information for these MAGs. These MAGs are derived from different samples. Could you please provide some data to support their distribution across different treatments? Do they belong to abundant groups or rare groups? For nitrogen cycling, genes related to N fixation process were not found in these genomes, what are the possible reasons? Was the phylogenetic tree in Supp Fig 12 constructed based on 16S sequences annotated from complete genomes? To what extent could these MAGs cover the isolated cultures?

Response: Thank you for your suggestions. We have provided the taxonomy of MAGs in the GitHub repository (<https://github.com/mxwang2016/Soybean>, L620-621 in the revision). For the abundance of MAGs, we draw their distribution pattern across treatments and attached here for your reference (see figure below), which showed less variation of MAGs among different treatments. Although it is kind of subjective to define abundant and rare microbial groups, we found that 68 out of 140 MAGs (48.6%) exhibited TPM value higher than 10, and 105 out of 140 MAGs (75.0%) exhibited TPM value higher than 5. These results, together with their relatively low variance among treatments and high consistency of functional genes related to N, P and K cycling in MAGs (Supplementary Fig. 10 in the revision), suggested that these MAGs were mainly composed of abundant groups with ubiquitous

distribution across samples. Among these MAGs, we did not find N fixation-related genes. This could be attributed to the (1) large microbial load and diversity in the rhizosphere masked the representative of N fixers, (2) relatively rare abundance but high variance of N fixation-related genes among treatments (evidenced in Fig. 6D in the revision), and/or (3) symbiotic N fixation outcompeting the commensal N-fixers.

The phylogenetic tree in Supplementary Fig. 12 (Supplementary Fig. 10 in the revision) was constructed based on the complete genomes assembled from metagenomic analysis. We have supplemented this information in the Methods part (L613-615 in the revision). Since our study on MAGs was mainly focused on the metabolic and functional potential in the rhizosphere microbiome, we did not conduct genomic sequencing of isolated bacterial strains. Considering most of isolated bacterial strains could be assigned as genus level in this study, we found that these MAGs covered 31.5% of genera from isolated cultures, indicating a relatively high representativeness of MAGs. Overall, our main objective of MAGs analysis is providing additional evidence in the common functional adaptations of rhizosphere microbiome under nutrient deficiency, by taking MAGs as examples of microbial genomes.

Figure. The distribution of MAGs among different treatments

L145-146: The statements seem to contradict the results in Fig 2D. In Fig 2, the potassium-deficient treatment but not the nitrogen-deficient treatment grouped separately.

Response: Sorry for the mistake about the information. We have replaced the Fig. 2D with the corrected form, which showed that the nitrogen-deficient treatment grouped separately from other treatments. We also double-checked the color labels in treatments of all figures across manuscript.

L273: Have these bacteria been identified at the species level? How many unique species were included? I believe it would be better to provide basic classification information for these 1011 bacterial strains. How many strains

performed >95% for the 16S rRNA gene with the same core ASVs? As I considered, each ASV can match multiple isolates. Did you randomly select one strain from the matched results as the SynCom member? Or what was the basis for this selection here?

Response: Thank you for your comments. We annotated these bacterial strains against the Greengenes database by 16S rRNA gene which we sequenced in this study. Overall, 248 unique sequences at species level were identified among 1011 strains (clustered at 97% identity), and 798 strains (78.9%) can be assigned as genus level, with totally 56 genera identified. Due to the limitation of 16S rRNA gene, only 234 strains (23.1%) can be assigned as species level based on Greengenes database. We have provided basic classification information of all bacterial strains in the GitHub repository (<https://github.com/mxwang2016/Soybean>, L620-621 in the revision) and supplemented related information in the revised manuscript (L282-283).

For the core ASVs, there were 55 strains showed high identity with core ASVs in three network modules at 95% similarity level. Although a core ASV could match with several isolated strains, we chose the strains with the highest identity to improve the representativeness. We have supplemented detailed information in the basis of selecting the bacterial strains for SynCom construction in the Methods part, following your suggestions (L550-555 in the revision).

In Fig 3A, does the term "relative abundance" refer to relative data derived from QMP? Or data from RMP?

Response: The term "relative abundance" is the relative data derived from RMP. Similar to Fig. 3, we also provided the accumulative absolute abundance of bacteria at phylum level by QMP in the Supplementary Fig. 6.

In Fig 6, does "feature number" refer to the number of ASVs?

Response: Yes, "the number of features" in the Y-axis of Fig. 6B (Fig. 7B in the revision) refers to the "number of ASVs". We have replaced "features" to "ASVs" to avoid the potential misunderstanding, and also double-checked the relevant descriptions across the manuscript to keep the consistency.

Reviewer #2 (Remarks to the Author):

The manuscript by Wang and colleagues is an interesting one, rich in results. This richness, however, can be of risk while sometime the authors wish to provide with as many results as possible without presenting/discussing them properly.

Some of the strong points of the analysis were considered as supplementary. For instance, the methodological comparison between QMP and RMP should be better presented at the top of Results section. The same applies to the functional (metagenomics) analysis which now is completely.

Response: We really appreciate the constructive suggestions. In this study, the main objective is to unveil the temporal dynamics of soybean root-associated bacteria and their functions in improving plant performance under unbalanced fertilization. We introduced the QMP method in high-throughput sequencing analysis, and provided new perspectives in the dynamic microbiome study. According to your suggestion, we have moved the results of QMP/RMP comparison and metagenomic analysis from supplementary figures to the figures in the main text, please see the Fig. 4 and Fig. 6 in the revision.

Anotehr element of confusion is the fact that some of the analyses were carried out on both endosphere and rhyzosphere, whereas others just on rhyzosphere (i.e, network analysis). Results are sometime only presented for a compartment, like the functional analysis only reported for rhyzosphere in Results while being done also on endosphere (see line 452 in Methods). explanations on the logic for these swings would be needed.

Response: Thank you for your helpful suggestions. We analyzed temporal dynamics of soybean in both rhizosphere and endosphere, and compared the sensitivity of these two compartments to unbalanced fertilization. We showed that the bacterial community in the rhizosphere was more sensitive to unbalanced fertilization than in the endosphere (Fig. 2, Supplementary Fig. 3 and Supplementary Fig. 7). Thereafter, we mainly focused on the taxonomic and functional changes of rhizosphere bacterial community to the unbalanced fertilization. Given the high diversity of rhizosphere bacterial community, we identified core ASVs in the rhizosphere and conducted network analysis to reveal the change of co-occurrence patterns of core bacterial taxa in each treatment, and validated the function of ecological clustered derived from the network.

In the previous version of manuscript, we did not show the results of network inference in the endosphere, but we attached it here for your reference (see figure below), which showed less responsive of network complexity across treatments in the endosphere, corresponding with the few changes of microbial composition in the endosphere (Supplementary Fig. 7). We have provided more related information for improving the logic flow of data analysis, please see the L213-214 and L263-266 in the revised manuscript.

Figure. Visualized co-occurrence networks in the root endosphere of each treatment.

While the part on SynCom is interesting I see this more as later add-on the authors gave included to further enrich manuscript. Indeed, methodology is quickly described and often difficult to follow. Also, the analysis on the plant growth was limited to just two measured parameters (shoot height and total biomass) at 2 weeks after germination. Authors may consider to not include this part and present in a separate, more detailed, publication. If they want to have it, then methodology for constructing the network to compare fertilization-induced changes in ecological clusters has to be better explained in both Results and Methods.

Response: We appreciate the reviewer's constructive suggestions. We have provided more detailed description on the principles of constructing synthetic bacterial community to make it easy to follow (L550-555 and L569-572 in the Methods part). In the SynCom experiment, we repeated the experiment up to 25 days with similar results. We measured nitrate nitrogen contents of soybean plants, which showed significantly higher contents in the plants after inoculating SynCom5 or SynCom7 compared with SynCtrl and Water treatments (Supplementary Fig. 14 in the revision). We also conducted microbial functional experiments to explain the plant growth-promoting effects of SynComs (Supplementary Fig. 15 in the revision).

Supplementary Fig. 14 Synthetic microbial communities promote plant growth.

Supplementary Fig. 15 Plant growth-promoting (PGP) functions of synthetic bacterial communities.

Finally, I found the Discussion poor compared to the amount of findings. An assessment of the differences found and the possible reasons behind between bulk soil, endosphere and rhizosphere in all the treatments is missing. Also, an evaluation of the advantages of QMP over RMP is lacking. Same for the functional (metagenomics) findings.

Response: Thanks for the detailed suggestions. We have supplemented related discussions towards methodological advantages of QMP in microbiome study, plant selective forces between compartments and environmental perturbations (i.e. nutrient deficiency), as well as importance of functional adaptation in the rhizosphere for host homeostasis in the Discussion part, following your suggestions. Please see the L347-350, L359-365 and L377-381 in the revised manuscript.

Other comments:

- it would be important to add hypothesis at the bottom of the Introduction. This may also help in figuring out what are key results to be presented and, thus, better structuring the following sections.

Response: Thank you for the suggestion. We have added the hypothesis in the Introduction part to read as “We hypothesize that (1) the QMP would reveal a distinctive dynamic pattern in root-associated microbiome assembly; (2) the dynamics of root-associated bacteria would be largely affected by the lack of N, P or K fertilizers; (3) functional adaptation of the rhizosphere bacteria under nutrient deficiency would benefit soybean growth” in L98-102 in the revision.

- lines 134-137: how you can say that that time corresponds to the onset of root nodulation? Please provide evidence/reference for that.

Response: Thank you for your suggestion. The onset of root nodulation was observed in the field during sampling, and we sequenced the rhizobial community (based on *rpoB* gene) in the nodule from the day 14 (see Supplementary Fig. 8 in the revision). We have rephrased the sentence to read as “which corresponds to the observed onset of root nodulation in the field at 14 days after germination” in L139-140 in the revision.

- lines 145-146: in fig 2D it seems -K not -N treatment that separates from others.

Response: Sorry for the mistake here. We have replaced the Fig. 2D to the corrected form, which indicated the -N treatment grouped separately from others. We also double-checked the color-marking in treatments of all figures across manuscript.

- how were the DNA-read samples normalised? What methodology (e.g., SRS) was applied?

Response: Thank you for your suggestions. The relative microbiome profiling method was employed by rarefying to 1030 sequences per sample in QIIME 2. We have provided rarefaction curves and related descriptions on sequence processing in the revision (Supplementary Fig. 17 and L499-500). For the quantitative microbiome profiling, the sequencing reads did not do normalization, and the absolute abundance of each ASV was calculated based on the known spike copy number in each sample (Supplementary Fig. 16).

- p-values are missing in all figures

Response: Sorry for the missing of *P*-values, we have added *P*-values in all figure legends and main text.

- Figure 2A: were samples significantly different? Add letters to show this.

Response: Thank you for your suggestions. The initial bacterial diversity between Control and each unbalanced fertilization treatment in the bulk soil was insignificant in Fig. 2A, we have added the symbol to make it more readable, please see the revised Fig. 2A in the revision.

- Figures 2G-H are very difficult to read, please find a better way to present them. Also, on the x-axis what is the difference between "temporal distance" and "days of germination"?

Response: Thank you for the helpful suggestion. The Fig. 2G represents the correlation between the change of temporal distance (that is, change of sampling day between each two samples, Δd) and the dissimilarities of bacterial community (that is, Bray-Curtis distance between each two samples) in each treatment in the rhizosphere and endosphere, respectively. In this way, the temporal distance (Δd) and Bray-Curtis dissimilarity from every pair of two samples can be calculated in each treatment. The linear regression slope of each treatment in Fig. 2G explained the dynamics and turnover of bacterial community with time, which suggested a significantly lower

bacterial turnover rate of -P treatment in the rhizosphere.

In contrast, the Fig. 2H reflects the deviation of bacterial community (Bray-Curtis dissimilarity) in each unbalanced fertilization treatment (i.e. -N, -P or -K treatment) to the Control in each sampling stage. The horizontal axis is the exact day of sampling (that is, days post soybean germination, d), whereas vertical axis is the Bray-Curtis distance between each unbalanced fertilization treatment (either -N, -P or -K treatment) and Control in each sampling time. Therefore, there were only three regression curves in each compartment, with eight points of time. The linear regression slope in Fig. 2H explained the dissimilarity between each unbalanced fertilization treatment and Control with plant growth, a sharp positive association represented unbalanced fertilization-induced effect on microbial community increased with plant developmental stage. We have updated the related results description and figure legend following your suggestions. Please see the L161 and L829-832 in the revised manuscript.

- Figure 3: are the significant differences among temporal stages for all the 4 treatments? Also, in the phyla barplots it is impossible to distinguish between treatments.

Response: Thank you for your comments. Yes, there were significant differences of microbial loads in both rhizosphere and root endosphere across plant developmental stages. The difference tests of microbial loads (i.e. bacteria and rhizobia) among temporal stages were conducted by combining all the data of four treatments in each stage. We have clarified the method for significance test in the figure legend and added symbols in the Fig. 3 to distinguish microbial composition from specific treatment in the revision (L840-841).

- Figure 3 refers to 16S data? and Supp Fig. 9 to *rpoB*? Please clearly indicate this in the legend.

Response: Yes, Fig. 3 is based on the 16S data, whereas Supplementary Fig. 9 (Supplementary Fig. 8 in the revision) is derived from the *rpoB* data. We have clarified them in the figure legend in the revision following your suggestion.

- Lines 335-337: not clear, to be rephrased.

Response: Thanks for your suggestion. We have rephrased the sentence to read as “Consistent with this, we observed that soybean root nodules were reduced in number and diameter, and rhizobial abundance in the rhizosphere was decreased under P deprivation.” in the revised manuscript (L357-359).

- Lines 495-508: how modules were generated? Please add a paragraph on top to explain the method. As it is, it is very hard to be understood.

Response: Thank you for detail comments. A network can be divided into different sub-communities, in which the nodes strongly correlated with each other and formed more clustered topology within a sub-community. These sub-communities, or modules, were recognized as ecological clusters representing specific functional groups. The Gephi platform is used to identify modules of microbial taxa in a network. We have added essential interpretations in the method for generating modules and the basis for SynCom selection to make it easier to be understood. Please see the revised manuscript (L532-537 and L550-555).

- in Results I would call it "soil" not "Mollisol".

Response: Thank you for the suggestion. We have replaced the “Mollisol” to “soil” across Results part, following your suggestion.

Reviewer #3 (Remarks to the Author):

Comment to Wang et al., Nature Communication NCOMMS-23-22973

Title: Dynamic root microbiome sustains soybean productivity over four-decades of unbalanced fertilization. The manuscript under consideration investigates temporal dynamics (days) of the root-associated bacterial community of soybean plants of a field trial with fertilization treatments (-N, -P, -K). The authors use metagenomics sequencing (16S rRNA and rpoB) and Co-occurrence network analysis to investigate the changes in specific bacterial groups across plant growth stages. In the last part, the authors selected and isolated three microbial communities from the field to test the plant growth-promoting functions in a pot experimental setup.

This is an extensive study presenting very detailed information on changes in microbial (bacterial communities) over time and during the growing season. The presentation is relatively clear and asks relevant and novel research questions. The experimental design is elegant, and the complex metagenomics results are well- presented and analyzed. However, the paper lacks suitable hypotheses and conclusions. The fact that the experiment was analyzed after a few decades of fertilization treatments needs more attention and analysis. In addition, more temporal dynamic analyses of the Co-occurrence bacterial network can improve the manuscript's conclusions and key messages. Finally, the verification experiment done by the authors is crucial and powerful. However, the results of specific communities of plant-promoting function need more connection to the nutrient status and specifically nitrogen status. The other nutrient limitation needs to be presented and experimented with older plants.

Below I mentioned a few aspects that can be improved in the manuscript to provide a more robust and holistic conclusion on the temporal resolution of plant-soil interaction.

Response: We really appreciate the reviewer's positive comments on the manuscript. We have made point-by-point response on these comments and made corresponding revision in the manuscript as below.

Major comments-

1. Long-term effect of fertilization – The four-decade effect of unbalanced fertilization needs proper attention, analysis, and discussion. Only one figure (fig. 1C) presents the soybean yield differences in 2017 and 2020; no long-term effect is presented. The slight reduction in yield is shown only in the -P treatment and not in the -N -K. This no reduction in yield raises questions about the relevance of the fertilization treatments. If the data is unavailable, the title and the abstract should be modified accordingly.

Response: Thank you for your comments. We showed that the soil chemical properties, with emphasis on soil fertility, were largely affected by the unbalanced fertilization, after four decades of crop plantation (three-year soybean-maize-wheat rotation). Although it is a snapshot of soil fertility over four decades, these results provide direct evidences for the long-term accumulative effect of fertilization on soil chemical properties. On the other hand, we collected the detailed data since the cooperation with our partners from several years ago, and the soybean yield showed similar results between two years. However, the soybean yield exhibited a different pattern with soil fertility, which raised the question and hypothesis regarding the microbiome-mediated promotion of soybean growth and nutrient supply. We have toned down the “long-term effect” related description in the title to read as “Dynamic root microbiome sustains soybean productivity under unbalanced fertilization”, following your suggestions.

2. Co-occurrence network temporal resolution – The temporal resolution of the Metagenomics sequencing is nicely presented, leading to interesting community structure differences among the fertilization treatments. However, in the co-occurrence network analysis, only an aggregation of the temporal resolution is analyzed (according to my understanding). I suggest dividing the time point into several stages (for example, early growth, mid-growth, and late) and analyzing the Co-occurrence network separately. Observing the temporal resolution of the bacterial

community (Fig. 2), It seems that the most dominant changes are in D14 and D42, and D60 (Shannon index Fig. 2C). In addition, analyzing the temporal co-occurrence network can provide more specific bacterial community to provide more robust results in the experimental validation stage.

Response: We appreciate the reviewer's suggestions. We have conducted the temporal resolution of bacterial co-occurrence networks in the rhizosphere by dividing the time point into three stages as reviewer suggested (that is, early growth stage: D1-D14; mid-growth stage: D28-D42; and late growth stage: D60-D72), in which ranged around two weeks of time span in sampling to make it comparable among stages (Supplementary Fig. 11 in the revision). We found a clear increasing trend of network complexity with plant development, with -N treatment consistently reaching to the highest network complexity (average degree) among treatments, and -K treatment sharply increasing network complexity from mid-growth to late growth stage. In fact, the aggregation of a network in each treatment (by combining all samples) reflects the potential interactions with plant development among core taxa, because sampling time is the only variable in shaping the co-occurrence pattern in each treatment. The more interconnected network in -N and -K treatments suggested a more clustered temporal associations in microbial taxa during plant development. For the keystone taxa in each network, we reanalyzed the keystone taxa based on *Zi-Pi* scores, and found the module hub was hardly observed, which suggested the microbial group rather a single taxon governed the function of microbial communities in the rhizosphere (Supplementary Fig. 12 in the revision). Therefore, the identification of key modules (ecological clusters) would be helpful for experimental validation. We have provided more descriptions for improving the logic of the results, please see L263-266 in the revised manuscript.

Supplementary Fig. 11 Visualized co-occurrence networks in the rhizosphere across difference stages of each treatment.

3. Keystone taxa – The idea of keystone taxa has become an important and encouraging research topic. Identifying a specific species of bacteria as keystone taxa can improve the paper's conclusions. See recent publications (Amit and Bashan "Top-down identification of keystone taxa in the microbiome" Nat Comm 2023; Banerjee et al., Keystone taxa as drivers of microbiome structure and functioning Nature Reviews 2018).

Response: Thank you for your suggestions. In the network of each treatment (Supplementary Fig. 12 in the revision), we aimed to reveal the general co-occurrence pattern of core bacterial taxa during plant growth stages

under different unbalanced fertilization regimes, since the time-induced change was the only variable in network construction. After identification of the keystone taxa by Z_i - P_i scores, we showed that none module hub was identified in each network, except for a module hub presented in the network of -P treatment (Supplementary Fig. 12C, see figure below). In this context, we hypothesize that the microbial keystone module, rather than keystone taxa, would be more important for functional attributes in the rhizosphere. Hence, we conducted the modularity analysis to compare the change of network modules among treatments (Fig. 7A). We also implemented more analysis in determining keystone taxa according to your suggestion, and showed similar results that a few of taxa was identified as “keystone”. We have provided more details for logic flow of network analysis and subsequent validation experiments following your suggestions, please see L263-266, L532-537 and L550-555 in the revised manuscript.

Supplementary Fig. 12C Identified keystone taxa of co-occurrence networks in the rhizosphere of each treatment.

4. Discussion about other organisms – The manuscript analyzes and discusses only the interaction between plants and bacteria. However, the effect of other organisms, such as fungi is not presented and this a weakness. Fungi, specifically arbuscular mycorrhiza, have been shown to influence the microbial community and the co-occurrence of the microbial network. Specifically, discussing the -P treatment requires the potential influent of nutrient supply by the fungi. Moreover, discussion about N and P limitations and the tripartite mutualisms (rhizobium:mycorrhizal:plant) can be discussed too (See Deveau et al., Bacterial–fungal interactions: ecology, mechanisms and challenges FEMS Microbiology 2018).

Response: Thanks for your insightful suggestions. Yes, the arbuscular mycorrhizal fungi played fundamental roles in facilitating plant P uptake. We have supplemented more related discussions about arbuscular mycorrhiza and root nodules in the revised manuscript and highlight research directions towards mutualistic relationship among rhizobium, mycorrhizal and plants. We have cited the article you suggested, as well as other relevant articles in the Discussion part. Please see L363-365 and L412-416 in the revised manuscript.

5. Validation experiment – Plants were measured two weeks after germination. More extended experiments (several weeks) can produce more robust results and effects on plant growth. In addition, a fertilization treatment to validate the exact microbial community to nutrient availability would be useful.

Response: Thanks for your suggestions. We retested plant phenotypes in a new experiment by inoculating SynComs with up to 25 days under N-free or N-amended conditions, respectively. We also measured plant nitrate nitrogen contents, other than plant height and dry weight. Similar to the results from two weeks, we found the positive effect of SynCom5 and SynCom7 on plant height and dry weight. Both SynCom5 and SynCom7 significantly increased the nitrate nitrogen concentrations compared with SynCtrl and Water treatments, which

suggested a robust plant beneficial effect of the low-N-enriched microbial cluster. In addition, when the planting time exceeded to a month, the plant was prone to lodging, possibly due to a small pot restricts the growth of soybean. The new data was included in Supplementary Fig. 14 (figure below) and L304-313 in the revised manuscript.

Supplementary Fig. 14 Synthetic microbial communities promote plant growth.

6. Title: the title does not really reflect the content (e.g. Dynamic root microbiome sustains soybean productivity over four-decades of unbalanced fertilization). The work is mainly on describing microbial communities in relation to fertilisation treatment and time after planting.

Response: Thanks for your suggestions. We have replaced the topic to “Dynamic root microbiome sustains soybean productivity under unbalanced fertilization”, following your suggestion. This topic contains two main contents of the manuscript: (1) dynamics of soybean root microbiome affected by different unbalanced fertilization treatments; and (2) key ecological clusters from dynamic rhizosphere microbiome (i.e. network analysis) promoted soybean growth.

7) A range of papers already showed microbiome dynamics in plants (or plant roots). The novelty of this work compared to other studies is unclear.

Response: Thanks for your comments. Uncovering the assembly of the root-associated microbiome is crucial for harnessing the potential of beneficial microbes in sustainable agroecosystem management. However, previous studies relying on relative microbial profiling (RMP) have limited our understanding of the quantitative development of root microbiomes and their dynamic interactions. By employing quantitative microbiome profiling (QMP), our study provides a first look at the quantitative assembly of the soybean root-associated bacteria across plant developmental stages, and unveils a causal relationship between a key microbial cluster and crop productivity. We have addressed the novelty and contribution of our work compared to microbiome dynamic with RMP in the Discussion part. Please see L320-328 in the revised manuscript.

Minor comments

Abstract

The word bacteria should be mentioned. Microbiome includes many organisms, however, the study is focused only on bacteria. In addition, the manuscript's conclusions mentioned in the abstract can be more focused and less general.

Response: Thank you for your suggestions. We have replaced the “Microbiome” by “Bacteria” to make it more specific, and rewrite the conclusion sentence to focus our main findings and implications in the Abstract (L38-39 in the revision).

What is meant with quantitative microbiome profiling (this is not really discussed in the methods, as far as I can see, but possibly I overlooked this).

Response: Thank you for your comments. The term “quantitative microbiome profiling” is an analogy of “relative microbiome profiling”, in which the relative abundance of specific taxa is the percentage of reads number assigning to this group to overall reads number in a given sample. The relative abundance-based method has been widely used in microbial sequencing study; however, it failed to unveil dynamic successions in a varied microbial load among samples. The “quantitative microbiome profiling” method is able to provide quantitative copy number of each taxon by spike-in targeted DNA fragments during PCR amplification procedure. In the revised manuscript, we compared the results derived from these two methods to further demonstrate the advances of QMP (see Fig. 4 in the revision). With the help of QMP, we illustrated the temporal dynamics of soybean root-associated microbiome and identified the low-N-enriched ecological cluster to test their function in plant growth promotion. We further emphasized the meaning and advantages of QMP in the Methods part in the revised manuscript, following your suggestion (L475-477 in the revision).

Introduction

Line 73: Starting the paragraph with “specifically” is strange

Response: Thank you for your suggestion. We have deleted this word in the revised manuscript.

Line 85: The term “microbial priming” should be mentioned (Bastida et al., Global ecological predictors of the soil priming effect” Nat Comm 2019; Oppenheimer-Shaanan “A dynamic rhizosphere interplay between tree roots and soil bacteria under drought stress“ elife 2023).

Response: Thank you for your suggestion. We have added the relevance information about “microbial priming” to read as “Access to these nutrients in plants can be bolstered by its rhizosphere microbiome, which was shown to specifically enrich mineral nutrient metabolism in comparison to bulk soil by microbial priming effects mediated with root exudates” in the revised manuscript (L83-85).

Line 97: General hypotheses are missing.

Response: Thank you for your comments. We have added the hypothesis to read as “We hypothesize that (1) the QMP would reveal a distinctive dynamic pattern in root-associated microbiome assembly; (2) the dynamics of root-associated bacteria would be largely affected by the lack of N, P or K fertilizers; (3) functional adaptation of the rhizosphere bacteria under nutrient deficiency would benefit soybean growth” in the revised manuscript (L98-102).

Methods

Line 390: More information about soil management is important such as irrigation, tillage, and cover crop.

Response: Thank you for your suggestion. The conventional tillage was adopted during the experimental season. Specifically, the ridge was established by a ridging plough, with ridge spacing of 0.65 m. The soybean seeds were sowed in two rows of each ridge. The field management included irrigation and weeding was adopted according to local farmer practices and was identical for the different treatments. After harvest, the field was fallow until the planting season of next year, with no cover crop planting. We have added more details in the revised manuscript (L428 and L431-436).

The sampling method requires more details. What is the percentage of sampled plants per plot? How does the number of sampled plants affect competition between the plants?

Response: Thank you for your comments. The cultivation density of soybean was around 250,000 plants per hectare, with plant spacing of 0.1 m in a row. For each plot, there are around 1,000 plants. In each sampling time, 3 plants of each plot were taken, with 24 plants in whole plant growth stage, comprising of only ~2.4% of total seedlings. For each plot, three plants with similar growth state were randomly chosen for collection, and the sampling location was apart from previous sampling stages to prevent the potential marginal effect and disturbance with adjacent plants, with 9 biological samples for each treatment at each stage. We have added the relevant information in the revised manuscript (L431-436 and L442-443), following your suggestion.

Line 415: What instrument was used to grind the samples?

Response: Thank you for your comments. The grinding instrument is a TissueLyser-48, which was produced by Shanghai Jingxin Industrial Development Co., Ltd. We have added the name and type of instruments in the revised manuscript (L462-463).

Line 450: The rarefaction curves can be useful in the supplementary information.

Response: Thank you for your suggestion. We have added the rarefaction curves as Supplementary Fig. 17 in the revised manuscript.

Supplementary Fig. 17 Rarefaction curves of bacterial Shannon index.

Line 490: Why a different set of primers was used?

Response: The primer pair 27F/1541R is used to target the full length of 16S rRNA gene for bacterial strain identification by Sanger sequencing, which includes the V5-V7 region of 16S rRNA gene that used for high-throughput sequencing.

Results

Fig. 2: The X-axis in panels 1B and 1C are misleading; the number of days between the time points is not equal (4 days, 3 days, 7 days, 17 days...). Changes happening at the beginning of the growth stage (days 1-7) are much quicker than the later stages (days 28-42).

Response: Thank you for your suggestions. We have replaced X-axis of Fig. 2B and 2C to the corrected form in the revised manuscript. We also double-checked the presentation of X-axis in other figures in the revision.

Line 121: This statement requires more experiments. The nitrogen amount is significantly less depleted compared to the other nutrients.

Response: Thank you for your suggestions. We have deleted this statement in the revised manuscript.

Figure 3: Why the -N treatment has such a high standard error (BS).

Response: Thanks for your comments. The absolute abundance of bacteria in the -N treatment of bulk soil was ranging from $1.03 - 2.98 \times 10^9$ copies g^{-1} , which was similar to other treatments and compartments. The observed “high standard error” in the -N treatment is mainly because of the low range of Y-axis (from 9.0 to 9.5) in Fig. 3A.

Line 174: The significant effect is in the rhizosphere and less in the endosphere. Analysis comparing the two can be important.

Response: Thank you for your suggestion. We have added the relevant information about “endosphere microbiome” to read as “By contrast, the change of bacterial loads between Control and -P treatment in the endosphere was hardly observed and not consistent during plant developmental stages (Fig. 3C)” in the revised manuscript (L179-180).

Fig S7: The figure is very nice and important. I will consider moving it as a main figure instead of Fig. 3. Please mention the different y-axis units in the different panels in the text.

Response: Thank you for your suggestion. The Fig. 3 presents the overall landscape of bacterial loads and composition, whereas the Supplementary Fig. 7 would contain more information in the direct comparison of temporal dynamics of bacterial phyla between QMP and RMP. Therefore, we would move the Supplementary Fig. 7 to the main text (Fig. 4 in the revision) and also retain the original Fig. 3. We have mentioned the different Y-axis units in the figure legend to make it easier to be read (L846-849 in the revision).

Line 190: Day 14 is a critical day with many changes. Why? What happened after two weeks? This may be relevant to the rhizobia establishment process.

Response: Thanks for your comments. Nodulation is an important event for soybean, and previous studies have demonstrated a substantial physiological change during organ establishment. We observed a common nodulation process at D14 in the field, which may affect the root-associated microbiome by competition or host repulsion. We have provided more related information in the revision (L139-140) for better explanation.

Line 190: I do not get the last point. Why the Bacteroidetes group is one of the most responsive to plant growth? Maybe to plant growth stages. The Proteobacteria show the same pattern.

Response: Sorry for the confusion here. Based on the change of bacterial phylum, we identified that the Bacteroidetes is the most responsive group to plant growth stages, followed by Proteobacteria. We have rewritten the sentence to read as “These results suggest that the Bacteroidetes and Proteobacteria are the most responsive groups to plant developmental stages” in L194-195 of the revised manuscript, following your suggestion.

Fig S8: At which time point is this date based?

Response: The results of Supplementary Fig. 8 (Supplementary Fig. 7 in the revision) were based on the all samples from different stages, based on the paired Wilcoxon test. Given the fact that the abundance of bacterial phylum substantially varied with plant developmental stages, we averaged the absolute abundance of each bacterial phyla from each treatment and stage, and employed paired Wilcoxon test between each unbalanced fertilization treatment and Control treatment, which would be helpful to find the consistence responsive taxa in each treatment, irrespective of the change of microbial abundance with time. We have added the relevant descriptions in the figure legend in the revised Supplementary Fig. 7.

Line 255: Belong to the discussion, and the results don't support this claim.

Response: Thank you for your suggestion. We have deleted this sentence in the revised manuscript.

Fig. S11: very nice figure. I will move the figure to the main part of the article.

Response: We appreciate for your positive comments on the results of metagenomic analysis. In this figure, we found bacterial functional genes related to nutrient supply were enriched in the unbalanced fertilization treatments to the Control, which was the additional support to our findings. We have moved it to the main part of the manuscript (Fig. 6 in the revision), following your suggestion.

Discussion

Line 315: "Bacteroidetes were particularly enriched, indicating that they were the most responsive to plant growth". Proteobacteria show the same pattern.

Response: Thank you for your suggestion. We have revised this sentence to read as "Bacteroidetes were particularly enriched indicating that they were the most responsive to plant growth, followed by Proteobacteria." in the revised manuscript (L334).

Line 337-341: All these results came from the same experiment? If yes, add in which exact figure.

Response: Yes, these results were come from the metagenomic analysis of microbial functional genes. We have supplemented the figure citations across the Discussion part, following your suggestion.

Line 329-348- The paragraph needs to be organized better. I assume you tried to discuss the effect of environmental parameters on the microbial community, but the structure needs to be written better.

Response: Thank you for your suggestion. We have separated them into two paragraphs and provided more discussions related to the linkage between microbial dynamics and functions related with plant phenotypes. Please see L358-365 and L377-381 in the revision.

Line 369: How and where was it shown that the module#1 effect is inconsistent and probably not rhizobia dependent effect?

Response: Sorry for the confusion here. We show that the SynComs derived from the low-nitrogen-enriched module (module#2) are capable of stimulating plant growth, without the inoculation of rhizobia. In contrast, the SynCom from module#1 does not support plant growth, with only marginal effect on the plant dry weight. Hence, we think the low-nitrogen-enriched module promotes plant growth in a rhizobia independent way. We have added a sentence to read as "We did not observe the formation of root nodules in SynCom treatments, suggesting that the effect was not dependent on the presence of rhizobia" for better explanation (L403-404 in the revision).

REVIEWERS' COMMENTS

Reviewer #1 (Remarks to the Author):

The authors have thoroughly addressed all the issues raised in the reviewer's comments. The revised manuscript has greatly improved. I believe that the current version meets the requirements for publication.

Reviewer #2 (Remarks to the Author):

The authors have taken into account previous suggestions and the overall quality of the manuscript has improved. I still think that some of the parts, especially in the Results section, are hard to follow as too wordy. I would try to shorten and better focus on main findings.

Minor comments:

L64: full name for N

L95: combined to what? Not "to investigate".

L124-125: I would avoid mentioning not significant results (-K treatment).

L145: vs. not v.s.

L204 and Fig. 3: Saccharibacteria instead of TM7?

L360-365: I would rewrite this. Your findings only support reason one, the other is an hypothesis but you have not proofs for that.

L415:"for benefiting plants through..."

Reviewers #3 and #4 (Remarks to the Author):

The study shows temporal dynamics of root-associated bacterial communities on soybean plants with -N, -P, -K fertilization treatments. The major caveats of the previous draft were the temporal dynamic analysis of the bacterial network, a link between the long-term fertilization treatments and the temporal bacteria community, and the verification experiment with other nutrient limitation treatments and in several growth stages.

All the comments mentioned by the reviewers improved significantly in the new version. The revised manuscript contains an additional analysis of co-occurrence networks of the temporal resolution, keystone taxa analysis, and further discussion about other organisms besides bacteria. The work is highly valuable and provides additional support for the dynamics of root-associated bacteria and the promoting plant growth potential. A few topics can get more attention and improve the manuscript in my opinion:

1. Long-term effect of fertilization –The long-term effect of the unbalanced fertilization presented in Fig. 1 is informative and well presented. However, further discussion and comparison of the yield effect to other long-term fertilization experiments worldwide are missing. Furthermore, general conclusions, including the link between the long-term field results and the bacterial community dynamic investigated in this project, can improve the final conclusions of the manuscript.

2. Validation experiment –The resubmitted manuscript includes another harvesting time point, which is nicely done! However, further connection to the field yields can improve the paper more. For example, to test the plant phenotype in the validation experiment at the same growth stages described in the first part of the manuscript. In addition, it's unclear why the authors decided to test only –N fertilization and not –P, which has significant results in the long-term unbalanced fertilization and specific P-related enzymes suggested to be involved. Validation experiment with other nutrient limitation treatments (–P and –K) and several growth stages can validate the proposed microbial dynamic mechanism more holistically.

3. The revised manuscript presents nicely the bacterial dynamic network in different growth stages (Fig. S11). Presenting the significant differences among the various growth stages can improve the analysis.

REVIEWERS' COMMENTS

Reviewer #1 (Remarks to the Author):

The authors have thoroughly addressed all the issues raised in the reviewer's comments. The revised manuscript has greatly improved. I believe that the current version meets the requirements for publication.

Response: We appreciate for your positive comments on the manuscript.

Reviewer #2 (Remarks to the Author):

The authors have taken into account previous suggestions and the overall quality of the manuscript has improved. I still think that some of the parts, especially in the Results section, are hard to follow as too wordy. I would try to shorten and better focus on main findings.

Response: We appreciate for your positive comments and suggestions. We have deleted some non-significant and meaningless descriptions in the Results section to shorten the manuscript, following your suggestions.

Minor comments:

L64: full name for N

Response: Thank you. We have added the full name of N.

L95: combined to what? Not "to investigate".

Response: Thank you. We have revised it.

L124-125: I would avoid mentioning not significant results (-K treatment).

Response: Thank you for your suggestion. We have deleted the statement of non-significant results.

L145: vs. not v.s.

Response: Thank you. We have corrected it across the manuscript.

L204 and Fig. 3: Saccharibacteria instead of TM7?

Response: Thank you. We have revised it across the manuscript.

L360-365: I would rewrite this. Your findings only support reason one, the other is an hypothesis but you have not proofs for that.

Response: Thank you for your suggestion. We have rewritten this sentence in the revised manuscript.

L415: "for benefiting plants through..."

Response: Thank you. It has been revised.

Reviewers #3 and #4 (Remarks to the Author):

Manuscript NCOMMS-23-22973, Wang et al., Revision-1

The study shows temporal dynamics of root-associated bacterial communities on soybean plants with -N, -P, -K fertilization treatments. The major caveats of the previous draft were the temporal dynamic analysis of the bacterial network, a link between the long-term fertilization treatments and the temporal bacteria community, and the

verification experiment with other nutrient limitation treatments and in several growth stages.

All the comments mentioned by the reviewers improved significantly in the new version. The revised manuscript contains an additional analysis of co-occurrence networks of the temporal resolution, keystone taxa analysis, and further discussion about other organisms besides bacteria. The work is highly valuable and provides additional support for the dynamics of root-associated bacteria and the promoting plant growth potential. A few topics can get more attention and improve the manuscript in my opinion:

Response: We appreciate for your positive comments and constructive suggestions. We have made point-by-point response on these comments and made corresponding revision in the revised manuscript.

1. Long-term effect of fertilization –The long-term effect of the unbalanced fertilization presented in Fig. 1 is informative and well presented. However, further discussion and comparison of the yield effect to other long-term fertilization experiments worldwide are missing. Furthermore, general conclusions, including the link between the long-term field results and the bacterial community dynamic investigated in this project, can improve the final conclusions of the manuscript.

Response: Thank you for your insightful suggestions. We have added more discussions related to other fertilization experiments globally and legacy effect of long-term fertilization on bacterial community in the revised manuscript, following your suggestions.

2. Validation experiment –The resubmitted manuscript includes another harvesting time point, which is nicely done! However, further connection to the field yields can improve the paper more. For example, to test the plant phenotype in the validation experiment at the same growth stages described in the first part of the manuscript. In addition, it's unclear why the authors decided to test only –N fertilization and not –P, which has significant results in the long-term unbalanced fertilization and specific P-related enzymes suggested to be involved. Validation experiment with other nutrient limitation treatments (–P and –K) and several growth stages can validate the proposed microbial dynamic mechanism more holistically.

Response: Thank you for your suggestions. In the network analysis, we showed that -N treatment significantly enriched the absolute abundance of bacterial taxa in Module#2, which suggested that the microbial group in Module#2 might associate with the plant phenotype in -N treatment. Therefore, we designed two SynComs from Module#2 for experimental validation. For the -P treatment, however, the absolute abundance of bacterial taxa in all of three modules, as well as the total bacterial load in the rhizosphere, were consistently depleted, suggesting that rhizosphere microbial taxa may reduce their functional potential in fully compensating plant phenotype in the field under low P condition. Therefore, we chose Module#2 to validate its plant growth-promoting functions, as Module#2 was the only cluster enriched in -N treatment, and taken Module#1 as the Control SynCom (no significant difference between Control and -N treatments). By the way, our results of the SynComs in Module#2 showing plant growth-promoting effects further provided clear evidences that the depleted microbial taxa in -P treatment contribute to plant fitness. Although a long-time monitoring on the plant phenotype by inoculating the low-nitrogen-enriched microbial cluster might contribute to the validation of the results, the pot experiment could restrict the soybean growth to mature. Therefore, further field experiment to test the yield effects of SynComs and their synergistic effect with rhizobia is our next plant in the near future, and the formulation of microbial inoculant and their survival rate will be also taken into consideration in the field experiment.

3. The revised manuscript presents nicely the bacterial dynamic network in different growth stages (Fig. S11). Presenting the significant differences among the various growth stages can improve the analysis.

Response: Thank you for your suggestion. We have supplemented the significance test in Fig. S11 in the revision.